# Beyond Instance-Level Self-Supervision in 3D Multi-Modal Medical Imaging

Tan Pan [1 2 3]  Shuhao Mei[⋆ 4]  Yixuan Sun [1 2]  Kaiyu Guo [5 2]  Chen Jiang[⋆ 1 2]  Zhaorui Tan [3]
Mengzhu Li [1]  Limei Han [1]  Xiang Zou [4]  Yuan Cheng[⋆ 1 2]  Mahsa Baktashmotlagh [5]

## Abstract

Self-supervised pre-training methods in medical imaging typically treat each individual as an isolated instance, learning representations through augmentation-based objectives or masked reconstruction. They often do not adequately capitalize on a key characteristic of physiological features: anatomical structures maintain consistent spatial relationships across individuals (instances), such as the thalamus being medial to the basal ganglia, regardless of variations in brain size, shape, or pathology. We propose leveraging this cross-instance topological consistency as a supervisory signal. The challenge arises from the inherent variability in medical imaging, which can differ significantly across instances and modalities. To tackle this, we focus on two alignment regimes. (i) Intra-instance: with pixel-level correspondences available, a cross-modal triplet objective explicitly preserves local neighborhood topology. (ii) Inter-instance: without such supervision, we derive pseudo-correspondences to control partial neighborhood alignment and prevent topology collapse across modalities. We validate our approach across 7 downstream multi-modal tasks, achieving average improvements of 1.1% and 5.94% in segmentation and classification tasks, respectively, and demonstrating significantly better robustness when modalities are missing at test time.

[1]Fudan University, China [2]Shanghai Academy of AI for Science, China [3]Bioinformatics Institute (BII), Agency for Science, Technology and Research (A*STAR), Singapore [4]Huashan Hospital, National Center for Neurological Disorders, Fudan University, China [5]University of Queensland, Australia. Correspondence to: Shuhao Mei <shmei21@m.fudan.edu.cn>, Chen Jiang <jiangchen@sais.org.cn>, Yuan Cheng <chengyuan@fudan.edu.cn>.

*Proceedings of the 43rd International Conference on Machine Learning*, Seoul, South Korea. PMLR 306, 2026. Copyright 2026 by the author(s).

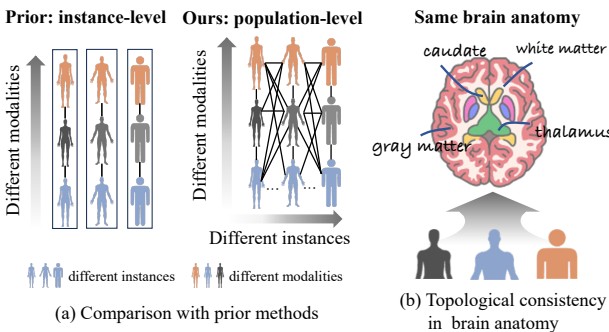

*Figure 1.* (a) Comparison with prior methods. Prior work learns instance-level representations via self-supervision, treating each individual independently. In contrast, our method learns cross-individual, multi-modal representations that capture relations across individuals. (b) Topological consistency. Human anatomy shares population-level topology (common structures and spatial layout), offering a natural prior for representation learning.

## 1. Introduction

Self-supervised learning (SSL) has become the dominant paradigm for visual representation learning, with contrastive learning (He et al., 2020), knowledge distillation (Grill et al., 2020), and masked autoencoding (He et al., 2022) achieving strong results across domains. In medical imaging, approaches are adapted with domain priors such as anatomical invariance (Pan et al., 2025; Jiang et al., 2023) and cross-modal consistency (Rui et al., 2025; Tan et al., 2025). Yet a common limitation remains: nearly all methods treat each individual as an independent instance, learning by contrasting augmented views or reconstructing masked regions within the same individual. While effective for ensuring robustness, those paradigms overlook a key characteristic of physiological features: human anatomy exhibits a stable, shared structure across individuals (instances). As a result, it underutilizes cross-instance regularities that could further enhance the quality of the learned representations.

Building on this, Fig. 1 (b) highlights the topological consistency observed in brain anatomy across different individuals. Despite considerable variations in brain size, shape, and pathology, the spatial relationships between anatomical structures remain remarkably stable. For example, the thalamus consistently lies medially to the basal ganglia, ventricles are adjacent to deep gray matter structures, and

cortical regions maintain their relative neighborhood patterns. This consistency, crucially, exists not at the pixel level but in the structural relationships between regions (Segobin et al., 2024). Previous work (He et al., 2023) leverages explicit geometric consistency to align inter-image representations in self-supervised learning, typically using known or learned transformation matrices (e.g., affine transforms). However, implicit transformations or discrepancies, such as those arising across modalities, are difficult to parameterize or recover with an explicit matrix, making such formulations less applicable in cross-modal or cross-instance scenarios.

To address this, we introduce TACO, a pre-training framework that leverages **T**opology-**A**ware **CO**nsistency across individuals as a form of multi-modal and multi-instance supervision. The key idea in this paper is to ensure that if two patches belong to the same region in scan $A$, their corresponding features from scan $B$ should also co-cluster as the same region in $B$'s representation space. This form of supervision goes beyond traditional instance-level learning: instead of focusing solely on the appearance of individual structures, the model learns the relative positioning of structures within their broader anatomical context, as shown in Fig. 1(a). To achieve this, we use a dual alignment approach: intra-instance and inter-instance consistency. For intra-instance alignment, we apply a cross-modal triplet objective, preserving local neighborhood topology within the same individual across modalities. For inter-instance alignment, we use unsupervised Mutual Nearest Neighbor (MNN) search to establish pseudo-correspondences, ensuring partial neighborhood alignment across instances.

We conduct extensive experiments across 7 tasks, demonstrating that our method achieves state-of-the-art (SOTA) performance and exhibits significantly better robustness in missing-modality scenarios, where some imaging sequences are unavailable at test time. To further validate our approach, we present t-SNE visualizations of the learned embedding space. The results show that our method effectively clusters feature tokens by anatomical identity across different individuals and modalities, which aligns well with our intended goal. In contrast, baseline self-supervised methods tend to cluster features by individual identity rather than anatomical correspondence, indicating that they capture instance-specific patterns rather than generalizable anatomical structures. Our main contributions are as follows:

(1) To the best of our knowledge, we are the first to jointly model inter-instance and inter-modality topological consistency as a supervisory signal for medical image pre-training;

(2) We develop TACO, a pre-training framework that integrates intra-instance objectives (cross-modal alignment) with inter-instance constraints (cross-instance topological alignment) through partial correspondence matching;

(3) We demonstrate consistent and significant improvements over SOTA medical SSL baselines across 7 downstream multi-modal tasks, including those with challenging missing-modality scenarios. Additionally, visualizations of region features provide compelling evidence of the effectiveness of our proposed method, further validating its potential for enhancing medical image pre-training. Code will be publicly available at https://github.com/Ashespt/TACO.

## 2. Related Work

**3D medical self-supervised learning.** Most 3D SSL work targets CT and MRI. Early approaches (Wang et al., 2023c; Hatamizadeh et al., 2021; Wald et al., 2025) follow reconstruction- or contrastive-based objectives, mirroring general-purpose SSL. More recent methods incorporate anatomical invariance to enforce geometric consistency (He et al., 2023; Pan et al., 2025). While these techniques deliver strong results, they typically ignore multi-modality settings and largely confine supervision to intra-instance (within-scan) structure.

**Pre-training in 3D multi-modal medical imaging.** Medical pretraining has emerged as a promising route to cross-modality generalization (Chen et al., 2020; Tang et al., 2022; Jiang et al., 2023; Wu et al., 2024; Tan et al., 2025). For example, (Jiang et al., 2023) captures class-level anatomical invariants but overlooks patient-specific cues; Brain-MVP (Rui et al., 2025) distills modality-aware templates; and PUIR (Tan et al., 2025) emphasizes personal-level information while maintaining cross-modal alignment. However, these approaches still tend to treat scans in isolation, limiting the use of cross-patient topology and inter-instance relations that are crucial for robust multi-modal representation learning.

**Missing modality segmentation.** Most current work that focuses on generalization across modalities introduces tasks such as modality transfer and missing modality segmentation using structural modalities (Flair, T1, T2, and T1ce) of MRI for brain tumor segmentation (Zhao et al., 2022). There are three main types of approaches to missing modality segmentation. Knowledge distillation-based approaches transfer knowledge from models with complete modality information (teachers) to models with missing modality information (students) (Chen et al., 2021; Wang et al., 2023b). mmFormer (Zhang et al., 2022) and RFNet(Ding et al., 2021) recover missing information by leveraging the multi-modal latent feature space. Domain adaptation-based methods aim to reduce the gap between models with complete and incomplete modalities by aligning their domains (Wang et al., 2021).

**Topology in medical imaging** Topological consistency is a natural and crucial property in medical images (Iglesias

et al., 2023; Fischl, 2012). TopoLoss (Hu et al., 2019) introduces a differentiable, topology-preserving loss and demonstrates its effectiveness for pathology segmentation. Subsequent topology-aware methods primarily target tubular structures (Wen et al., 2025; Elsayed et al., 2025). However, very few works extend topological consistency to the unsupervised/self-supervised setting, despite the fact that human anatomy exhibits an invariant topological structure across patients.

## 3. Preliminary

**Notation.** The data set comprises medical images collected from a set of distinct individuals $H$, including multiple modalities $M$ in each individual $h \in H$. Thus, we denote the dataset as $X = \{x_m^h | h \in H, m \in M\}$, where $x_m^h$ is the image of the individual $h$ in modality $m$. Given the transformer encoder $\mathcal{E}$, the corresponding output tokens are $Z_m^h = \{z_{m,k}^h\}_{k=1}^K = \mathcal{E}(x_m^h)$, where $K$ is the token number.

For every input $x_m^h$, we exploit the standard autoencoder pipeline with encoder $\mathcal{E}$ and decoder $\mathcal{D}$ as:

$$\hat{x}_m^h = \mathcal{D}(\mathcal{E}(x_m^h)). \tag{1}$$

where, $\hat{x}_m^h$ is the reconstructed image. The reconstruction objective is

$$\mathcal{L}_{uni} = ||\hat{x}_m^h - x_m^h||_2^2. \tag{2}$$

**Instance-agnostic Multi-modal Neighborhood Ranking Consistency.** Biological organs exhibit stable anatomical structure, with consistent spatial relationships observed across modalities and individuals. This consistency motivates us to learn representations that are invariant with respect to modality and instance, while being governed by spatial relationships. To this end, we propose the Instance-agnostic Cross-Modal Neighborhood Ranking Consistency (IM-NRC) principle for medical multi-modal alignment. Specifically, given two modalities, $i$ and $j$, corresponding to individuals $h$ and $g$ respectively, we assume the existence of a mapping $C_{i \to j}^{h \to g} : Z_i^h \to Z_j^g$ that establishes the spatial correspondence between $Z_i^h$ and $Z_j^g$. Then, the model should thus adhere to the following assumption:

**Assumption 3.1** (IM-NRC). Given patch token $z_{i,k}^h \in Z_i^h$, if $z_{i,k}^h$ is located in decision region $R_{i,h}^\alpha = \{z_i^h \mid \sigma_{i,h}(z_i^h) = \alpha, z_i^h \in Z_i^h\}$, then $z_{j,k}^g = C_{i \to j}^{h \to g}(z_{i,k}^h)$ should be in region $\{C_{i \to j}^{h \to g}(z_i^h) \mid z_i^h \in R_{i,h}^\alpha\} \subseteq R_{j,g}^\alpha = \{z_j^g \mid \sigma_{j,g}(z_j^g) = \alpha, z_j^g \in Z_j^g\}$, where $\sigma_{i,h}$ and $\sigma_{j,g}$ are classifiers or discriminators.

In the unsupervised setting, we cannot explicitly obtain $\sigma_{i,h}$ and $\sigma_{j,g}$. Thus, unsupervised methods, such as nearest neighbors, can be leveraged to estimate the decision

regions. Under the IM-NRC assumption, if $\mathcal{P}_{i,h}(k)$ represents the set of indices of neighborhood points for $z_{i,k}^h$, then $\{o \mid z_{j,o}^g = C_{i \to j}^{h \to g}(z_{i,l}^h), l \in \mathcal{P}_{i,h}(k)\}$ should correspond to the set of indices of neighborhood points for $z_{j,k}^g$. The same assumption applies to the non-neighborhood region. This indicates that the optimal distributions of the tokens from $Z_i^h$ and $Z_j^g$ should remain consistent semantics and geometry under the correspondence $C_{i \to j}^{h \to g}$. Therefore, to regularize the feature distribution across instances and modalities, we exploit the contrastive metric learning method, such as triplet loss, to constrain the semantic and geometrical consistency, where the triplet in $Z_j^g$ is defined by the geometry in tokens $Z_i^h$ under the correspondence $C_{i \to j}^{h \to g}$.

## 4. Method

This paper aims to learn modality-invariant, population-level representations for medical data that preserve the intrinsic local topology of each individual while ensuring cross-modal topological consistency. To achieve this, we propose intra- and inter-instance cross-modal alignment methods as shown in Fig. 2. In our approach, $\mathcal{L}_{\text{intra}}$ and $\mathcal{L}_{\text{inter}}$ are employed to regularize cross-modality and cross-instance features in a self-supervised manner by maintaining neighborhood-ranking consistency and enforcing Assumption 3.1. Given the reconstruction objective (2), our overall training objective can be formulated as

$$\mathcal{L}_{\text{total}} = \mathcal{L}_{\text{uni}} + \mathcal{L}_{\text{intra}} + \mathcal{L}_{\text{inter}}, \tag{3}$$

where $\mathcal{L}_{\text{intra}}$ and $\mathcal{L}_{\text{inter}}$ will be introduced in the following part of this section. Specifically, the intra-instance loss $\mathcal{L}_{intra}$ is described in Section 4.1, while the inter-instance loss $\mathcal{L}_{inter}$ is outlined in Section 4.2.

### 4.1. Intra-instance Constraint

Given data $x_i^h$ and $x_j^h$ from individual $h$, we have the latent features $Z_i^h = \mathcal{E}(x_i^h)$ and $Z_j^h = \mathcal{E}(x_j^h)$. Following the geometric equivalence assumption (Mao et al., 2022), we assume that correspondences among latent token features mirror the patch-level correspondences in the image domain. Let $C_{i \to j}^h$ denotes the predefined token-level correspondence that aligns the representations $Z_i^h$ and $Z_j^h$ for individual $h$. Specifically, as the shapes and spatial locations of structures maintain alignment across modalities for the same individual, we take $C_{i \to j}^h$ to be the identity on positional indices.

The key insight behind intra-instance cross-modal alignment is that local neighborhoods must be preserved across modalities under the correspondence $C_{i \to j}^h$. This implies that if tokens are close in $Z_i^h$, their corresponding tokens should also remain close in $Z_j^h$, and similarly, if tokens are far apart in $Z_i^h$, their corresponding tokens should remain distant in

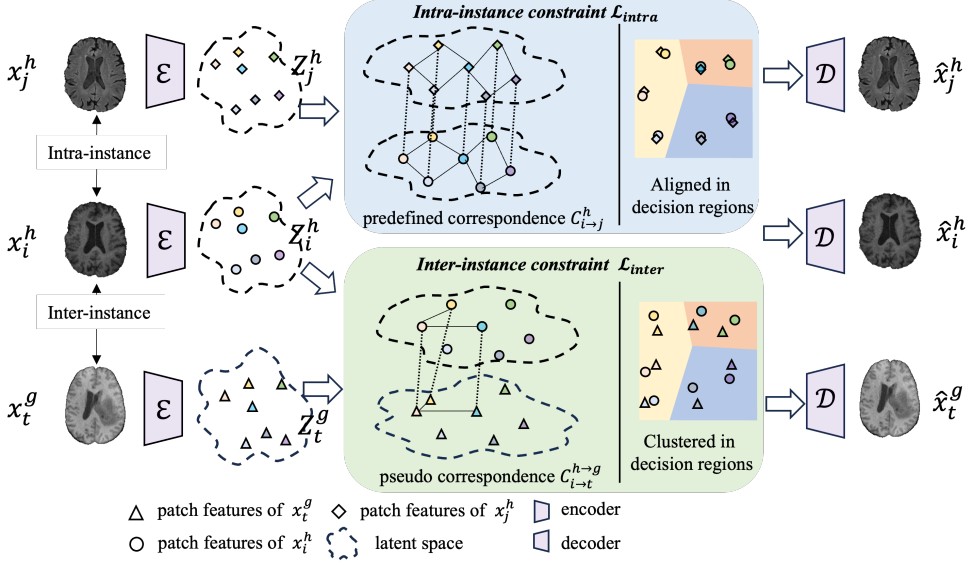

*Figure 2.* Pipeline of the proposed method. For each scan, a reconstruction loss learns unimodal representations. Across intra- and inter-individual pairs, following IM-NRC (Assumption 3.1), we align patch features of homologous anatomy, enforcing modality- and subject-invariant neighborhoods.

$Z_j^h$. To this end, we first obtain the pairwise distances in modality $i$ on the feature level, which can be denoted as matrix $D_i^h$. Then, the neighborhood set $\mathcal{P}_{i,h}^\omega(a)$ of the $a$-th patch in $Z_i^h$ can be defined as follows

$$\mathcal{P}_{i,h}^\omega(a) := \Big\{o\,\Big|\,D_i[a,o] \in Top_\omega(D_i[a,:]), o \neq a, o \leq K\Big\}, \tag{4}$$

where $Top_\omega(v)$ means the set of top-$\omega$ smallest values in vector $v$.

Given the correspondence $C_{i \to j}^h$, the set of positive features for $z_{j,a'}^h = C_{i \to j}^h(z_{i,a}^h)$ in modality $j$ can be denoted as $\mathcal{P'}_{i \to j,h}^\omega(a) := \{o' \mid z_{j,o'}^h = C_{i \to j}^h(z_{i,o}^h), o \in \mathcal{P}_{i,h}^\omega(a)\}$, where the indices $o$ are from the neighborhood set of the $a$-th patch in modality $i$.

Additionally, we define the negative set of $a$-th patch in $Z_i^h$ as $\mathcal{N}_{i,h}(a) := \{o \mid o \leq K, o \notin \mathcal{P}_{i,h}^\omega(a), o \neq a\}$, which includes all patches except the ones in the neighborhood and the patch itself. In modality $j$, the corresponding negative set is $\mathcal{N'}_{i \to j,h}(a) := \{o' \mid o' = C_{i \to j}^h(o), o \in \mathcal{N}_{i,h}(a)\}$.

Hence, for a given $p$-th patch feature in $Z_i^h$, we can construct triplet sets in modality $j$ for contrastive loss as follows

$$\text{Tri}_{i \to j, a, h}^{\text{intra}} := \Big\{(a', p', n') \,\Big|\, a' = C_{i \to j}^h(a),$$
$$(a', n') \in \mathcal{P'}_{i \to j, h}^\omega(a) \times \mathcal{N'}_{i \to j, h}^\omega(a)\Big\}, \tag{5}$$

where $\mathcal{N'}_{i \to j,h}^\omega(a) \subset \mathcal{N'}_{i \to j,h}(a)$ is a randomly selected subset with $\omega$ samples (paired with top-$\omega$ positives). In the triplet set, $(a', p')$ is the positive pair and $(a', n')$ is the negative pair. Hence, to regularize the feature distribution in

$Z_j^h$ with the relation in $Z_i^h$, we propose the following loss that aligns with Principle 3.1,

$$\mathcal{L}_{\text{intra}}^{i \to j} = \frac{1}{|H|\,K\,\omega} \sum_{h \in H} \sum_{p \in Z_i^h} \sum_{\substack{(a',p',n') \\ \in \text{Tri}_{i \to j, a, h}^{\text{intra}}}}$$
$$\Big[d(z_{j,a'}^h, z_{j,p'}^h) - d(z_{j,a'}^h, z_{j,n'}^h) + \delta\Big]_+, \tag{6}$$

where $[\cdot]_+ = max(0, \cdot)$. Then, the final intra-instance loss is

$$\mathcal{L}_{intra} = \frac{1}{|M|(|M| - 1)} \sum_{i,j \in M} \mathcal{L}_{intra}^{i \to j}. \tag{7}$$

### 4.2. Inter-instance Constraint

The intra-instance loss described above is built on a prior one-to-one correspondence between tokens across modalities. However, in cross-subject settings, inter-individual anatomical and functional variability makes dense spatial alignment challenging. To overcome this limitation, we introduce a pseudo-correspondence for inter-instance constraints. This approach enforces local topological consistency only on partially matched token subsets, which are identified in an unsupervised manner as shown in Fig. 3.

Given two individual instances $h \in H$ and $g \in H$, let $Z_i^h$ and $Z_t^g$ denote the feature representations of $h \in H$ and $g \in H$ in modality $i$ and $t$ respectively. The pseudo correspondence between different instances is built based on mutual nearest neighborhood. Specifically, we define the correspondence $C_{i \to t}^{h \to g}$ under the distance metric $d(\cdot, \cdot)$ as

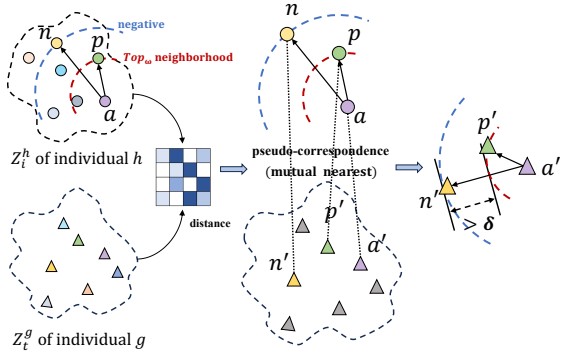

*Figure 3.* Inter-instance constraint. $\mathcal{L}_{inter}$ builds pseudo-correspondence by mutual-nearest-neighbor (MNN) and Top-$n$ neighborhoods across patients, enforcing modality-invariant clustering within the same decision region. Intra-instance constraint can be a special case of Inter-instance constraint, where $h = g$.

follows

$$C_{i \to t}^{h \to g}(z_{i,a}^h) = Z_{t,a'}^g \iff$$
$$a = \arg\min_o d(z_{i,o}^h, Z_{t,a'}^g) \wedge a' = \arg\min_o d(z_{i,a}^h, Z_{t,o}^g). \tag{8}$$

Then, we can build a set of solid indices according to the condition of the correspondence

$$V_{i,t}^{h,g} := \left\{ (a, a') \,\middle|\, a = \arg\min_o d(z_{i,o}^h, Z_{t,a'}^g), \right.$$
$$\left. a' = \arg\min_o d(z_{i,a}^h, Z_{t,o}^g) \right\}. \tag{9}$$

Given the $a$-th patch token feature $z_{i,a}^h \in Z_i^h$, similar to the Equation 5, we can define the triplet set for inter-instance loss

$$\text{Tri}_{i \to t, a, h \to g}^{\text{inter}} := \left\{ (a', p', q') \,\middle|\, (a, a') \in V_{i,t}^{h,g}, \right.$$
$$\left. (p', n') \in \mathcal{P}_{i \to t, h \to g}'^{\omega}(a) \times \mathcal{N}_{i \to t, h \to g}'^{\omega}(a) \right\}. \tag{10}$$

Thus, the inter-instance loss in $Z_t^g$ with $a$-th patch in $Z_i^h$ can be formulated as

$$\mathcal{L}_{\text{inter}}^{i \to t, h \to g}(a) = \frac{1}{\left| \text{Tri}_{i \to t, a, h \to g}^{\text{inter}} \right|} \sum_{(a',p',n') \in \text{Tri}_{i \to t, a, h \to g}^{\text{inter}}}$$
$$\left[ d(Z_{t,a'}^g, z_{t,p'}^g) - d(Z_{t,a'}^g, z_{t,n'}^g) + \delta \right]_+ . \tag{11}$$

Then, the loss for all patches is

$$\mathcal{L}_{inter}^{i \to t, h \to g} = \frac{1}{K} \sum_{a \in Z_i^h} \mathcal{L}_{inter}^{i \to t, h \to g}(a). \tag{12}$$

Then, the final inter-instance loss is

$$L_{inter} = \frac{1}{|M||H|(|M|-1)(|H|-1)} \sum_{i,t \in M} \sum_{h,g \in H} \mathcal{L}_{inter}^{i \to t, h \to g}. \tag{13}$$

Note that, if $V_{i,t}^{h,g} = \emptyset$, $L_{inter}^{i \to t, h \to g}$ will be set zero.

## 5. Experiments

Our experiments include evaluations on 4 segmentation tasks and 2 classification tasks. Beyond regular comparison with current SOTA medical SSL methods, we also compare missing modality segmentation ability with current SOTA missing-modality methods.

### 5.1. Experimental Setup

**Pre-training Datasets.** Following the multi-modal MRI pre-training protocol of BrainMVP (Rui et al., 2025), our model is pretrained on 16,022 3D scans from 3,755 patients. The corpus covers 8 modalities; for each patient, all modalities are registered before pre-training. As in BrainMVP, no segmentation annotations are used, and the pre-training data are disjoint from the downstream test sets. Details are shown in Tab 2, and additional details are provided in the Appendix.

**Downstream datasets.** Our downstream evaluation can be divided into three parts as shown in the Appendix Tab 2: 4 segmentation tasks (BraTS2023-PED (Kazerooni et al., 2024), BraTS2023-MET (Moawad et al., 2024), UPENN-GBM (Moawad et al., 2024)), and ISLES22 (Hernandez Petzsche et al., 2022), 2 classification tasks (ADNI (Jack Jr et al., 2008) and ADHD-200 (consortium, 2012)), and 1 missing modalities task (BraTS2018 segmentaion (Menze et al., 2014)).

**Implementation Details.** Following prior medical SSL works (Wu et al., 2024; Wang et al., 2023c; Pan et al., 2025; Tan et al., 2025), we use Swin-B (Hatamizadeh et al., 2021) as the pre-training backbone. We reproduce all pre-training methods in Tab. 1 on the same pre-training dataset; the only exception is BrainMVP, for which we use the publicly released checkpoint pretrained on the same dataset. Pre-training runs for 10k iterations with batch size = 2, optimized by AdamW (Loshchilov & Hutter, 2017) with weight decay = 1e-5 and learning rate = 3e-4. The distance function $d(\cdot, \cdot)$ is the cosine distance. All pre-training and fine-tuning are conducted on a single NVIDIA A100. For downstream tasks, we select the checkpoint with the best validation performance and report results on the test set. Additional details are provided in the Appendix.

### 5.2. Evaluation on Downstream Tasks

#### 5.2.1. COMPARISON WITH SOTA PRE-TRAINING METHODS

Since prior studies report that general-purpose SSL underperforms medical SSL on medical imaging, we restrict comparisons to medical pre-training baselines. For uni-modal methods, we include SwinMM (Wang et al., 2023c), Swin

*Table 1.* Results on four segmentation benchmarks. All methods are reproduced on the same pre-training dataset; for BrainMVP, we use its publicly released checkpoint trained on the same data. Metrics: TC (Tumor Core), ET (Enhancing Tumor), WT (Whole Tumor). **Best** and second best are highlighted. Our method achieves the best average performance.

| Pre-training Method | Backbone | BraTS2023-PED | | | | BraTS-MET | | | | UPENN-GBM | | | | ISLES22 | All Dataset |
|---|---|---|---|---|---|---|---|---|---|---|---|---|---|---|---|
| | | ET | TC | WT | avg. | ET | TC | WT | avg. | ET | TC | WT | avg. | IS | avg. |
| *From Scratch* | | | | | | | | | | | | | | | |
| - | UNET3D | 46.12 | 84.92 | 86.63 | 72.55 | 53.47 | 58.24 | 63.97 | 58.56 | 80.36 | 87.14 | 85.04 | 84.18 | 77.81 | 73.28 |
| - | UNETR | 46.97 | 78.02 | 81.73 | 68.9 | 51.42 | 54.03 | 60.16 | 55.2 | 79.77 | 84.5 | 84.78 | 83.02 | 76.51 | 70.91 |
| - | UniFormer | 47.81 | 82.34 | 83.82 | 71.32 | 59.49 | 62.42 | 64.77 | 62.23 | 80.43 | 86.62 | 85.37 | 84.14 | 76.52 | 73.55 |
| - | Swin UNETR | 51.11 | 86.86 | 88.15 | 75.37 | 62.1 | 66.07 | 68.31 | 65.5 | 82.33 | 86.7 | 84.18 | 84.02 | 77.33 | 75.56 |
| *Medical SSL* | | | | | | | | | | | | | | | |
| SwinMM (Wang et al., 2023c) | Swin UNETR | 51.99 | 84.71 | 85.55 | 74.08 | 64.26 | 68.13 | 68.47 | 66.95 | 82.53 | 88.02 | 85.69 | 85.41 | 77.95 | 76.10 |
| Swin UNETR (Tang et al., 2022) | Swin UNETR | **52.55** | 84.75 | 86.28 | 74.53 | 63.32 | 67.5 | 69.42 | 66.74 | 81.84 | 87.36 | 85.91 | 85.04 | 78.71 | 76.26 |
| VoCo (Wu et al., 2024) | Swin UNETR | 50.52 | 84.22 | 85.91 | 73.55 | 60.51 | 64.13 | 66.01 | 63.55 | 81.92 | 87.7 | **86.02** | 85.21 | 79.65 | 75.49 |
| M³AE (Liu et al., 2023) | UNET3D | 47.57 | 84.62 | 85.78 | 72.66 | 57.73 | 62.97 | 67.31 | 62.67 | 81.98 | **88.11** | 85.68 | 85.26 | 79.13 | 74.93 |
| BrainMVP (Rui et al., 2025) | UniFormer | 49.89 | 85.25 | 87.52 | 74.22 | 62.42 | 67.63 | 70.45 | 66.83 | 81.81 | 87.65 | 84.67 | 84.71 | 79.69 | 76.36 |
| S²DC (Pan et al., 2025) | Swin UNETR | 51.04 | 87.46 | 88.64 | 75.71 | 62.41 | 66.01 | 67.88 | 65.43 | 81.38 | 87.49 | 85.77 | 84.9 | 78.49 | 76.13 |
| **TACO** | Swin UNETR | 50.58 | **88.69** | **89.38** | **76.22** | **64.41** | **69.52** | **70.6** | **68.17** | **82.58** | 87.84 | 85.98 | **85.62** | 79.85 | **77.47** |

*Table 2.* The pre-training and downstream datasets in our work. Seg.: segmentation; cls: classification; ms: missing modality.

| Dataset | Task Type | Modality type | Cases |
|---|---|---|---|
| *Pre-training* | | 8 modalities | 3755 |
| BraTS2021 (Baid et al., 2021) | - | T1,T1CE,T2,FLAIR | 1470 |
| BraTS2023-SSA (Adewole et al., 2023) | - | T1,T1CE,T2,FLAIR | 75 |
| BraTS2023-MEN (LaBella et al., 2023) | - | T1,T1CE,T2,FLAIR | 1141 |
| UCSF-PDGM (Calabrese et al., 2022) | - | T1,T1CE,T2,FLAIR,DWI,ADC | 501 |
| IXI (Chen et al., 2022) | - | T1,T2,MRA,PD | 568 |
| *Downstream* | | 6 modalities | |
| BraTS2023-PED (Kazerooni et al., 2024) | seg. | T1,T1CE,T2,FLAIR | 99 |
| BraTS2023-MET (Moawad et al., 2024) | seg. | T1,T1CE,T2,FLAIR | 238 |
| UPENN-GBM (Moawad et al., 2024) | seg. | T1,T1CE,FLAIR | 127 |
| ISLES22 (Hernandez Petzsche et al., 2022) | seg. | FLAIR,DWI,ADC | 238 |
| BraTS2018 | seg. w/ ms. | T1,T1CE,T2,FLAIR | 1470 |
| ADNI (Jack Jr et al., 2008) | cls. | T1 | 697 |
| ADHD-200 (consortium, 2012) | cls. | T1 | 767 |

*Table 3.* Results on two classification benchmarks. All pre-training methods use the same pretrained model as in Tab. 1. **Best** and second best are highlighted. Our method achieves the best average performance in all metrics.

| Pre-training Method | Backbone | ADNI | | | ADHD | | | avg. | | |
|---|---|---|---|---|---|---|---|---|---|---|
| | | ACC | AUC | F1 | ACC | AUC | F1 | ACC | AUC | F1 |
| **From scratch** | | | | | | | | | | |
| - | UNET3D | 81.43 | 63.91 | 73.09 | 63.64 | 55.57 | 38.89 | 72.535 | 59.74 | 55.99 |
| - | UniFormer | 85.24 | 86.59 | 65.41 | 61.9 | 59.22 | 52.9 | 73.57 | 72.905 | 59.155 |
| - | Swin-B | 93.33 | 93.85 | 87.42 | 67.87 | 64.94 | 56.85 | 80.6 | 79.395 | 72.135 |
| **Medical SSL** | | | | | | | | | | |
| SwinMM | Swin-B | 89.05 | 90.87 | 77.32 | 52.81 | 53.39 | 53.7 | 70.93 | 72.13 | 65.51 |
| Swin UNETR | Swin-B | 94.2 | 93.7 | 93.47 | 65.37 | 70.59 | 62.39 | 79.79 | 82.15 | 77.93 |
| VoCo | Swin-B | 85.51 | 90.3 | 82.23 | 53.68 | 52.43 | 48.06 | 69.60 | 71.37 | 65.15 |
| M³AE | Unet3D | 83.33 | 83.76 | 66.44 | 61.47 | 57.33 | 60.05 | 72.4 | 70.55 | 63.7 |
| BrainMVP | UniFormer | 93.33 | 95.05 | 87.1 | **69.26** | 68.72 | 58.41 | 81.3 | 81.89 | 72.76 |
| S²DC | Swin-B | 88.57 | 95.2 | 76.67 | 64.94 | 70.34 | 63.06 | 76.76 | 82.77 | 69.87 |
| TACO | Swin-B | **96.19** | **97.09** | **93.57** | 67.53 | **71.14** | **63.82** | **81.86** | **84.12** | **78.7** |

UNETR (Tang et al., 2022), VoCo (Wu et al., 2024), and S²DC (Pan et al., 2025). For multi-modal pre-training, we compare against M³AE (Liu et al., 2023) and Brain-MVP (Rui et al., 2025).

**Results on segmentation tasks.** Tab. 1 compares training from scratch with medical SSL pre-training across four benchmarks. The Dice similarity coefficient is adopted as the metric for evaluation. Training from scratch shows clear gaps to SSL in most tasks, e.g., the best scratch baseline (Swin UNETR) reaches a Dice coefficient 65.5% on BraTS-MET, 84.02% on UPENN-GBM, and 77.33% on ISLES22, below SSL counterparts. Our method, built on Swin UNETR backbone, delivers the best average on ev-

ery dataset and the best overall average across all datasets: 77.47% Dice, surpassing the strongest medical pre-training baseline (BrainMVP, 76.36%) by **1.11%** points. While some baselines win isolated sub-metrics (e.g., SwinMM on BraTS2023-PED ET, M³AE on UPENN-GBM TC), our approach yields consistent gains on the dataset-level averages, suggesting that the proposed pre-training better preserves transferable structure across cohorts and tasks.

**Results on classification tasks.** Tab. 3 reports accuracy (ACC), AUC, and F1 score on ADNI and ADHD datasets. Training from scratch is clearly inferior, particularly on the complex ADNI dataset. Incorporating medical SSL pre-training methods yields substantial improvements across various backbones, with BrainMVP and S²DC emerging as the strongest baselines, achieving best average scores of 81.3%/82.77%/69.87%, respectively. TACO establishes a new state of the art on both datasets, and all averaged metrics, reaching 81.86%/84.12%/78.7%, which yields gains of **+0.56%/+1.35%/+5.94%** over the best baseline. These consistent gains across datasets and metrics indicate that the proposed pre-training transfers better to classification than prior medical SSL methods, especially in terms of decision quality (F1) while maintaining strong discrimination (AUC).

### 5.2.2. MISSING MODALITY SEGMENTATION

**Missing-modality setup.** We follow the evaluation protocol of recent SOTA methods: PUIR (Tan et al., 2025), mmFormer (Zhang et al., 2022), SPA (Wang et al., 2023a), M³AE (Liu et al., 2023), and M2F (Shi et al., 2023). Following PUIR, we use the BraTS23 (Kazerooni et al., 2024) multi-modal brain tumor dataset, which provides four structural MRI modalities per subject (T1, T1ce, T2, FLAIR). We pretrain the models on BraTS23 and then fine-tune them on BraTS18 for segmentation, mirroring PUIR's protocol; BraTS18 contains the same four modalities. The baseline results are from PUIR (Tan et al., 2025). More details can be found in the Appendix.

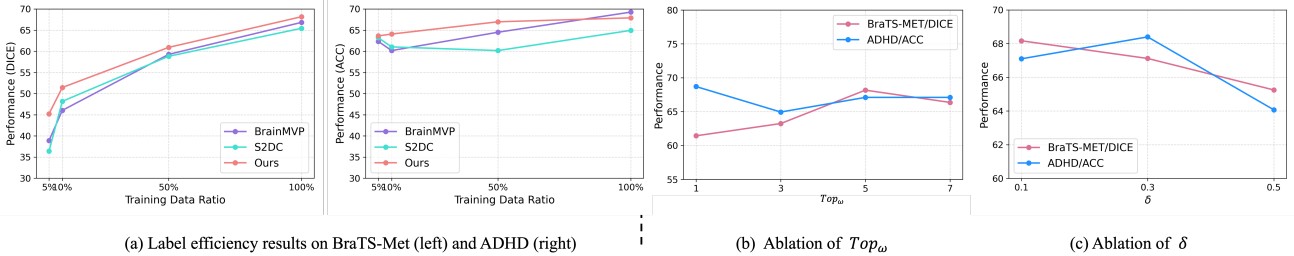

*Figure 4.* (a) Label efficiency results of the downstream segmentation task BraTS-MET and the classification task ADHD. Our method shows robust gains across data scales and tasks. (b) Ablation results of $Top_\omega$. (c) Ablation results of $\delta$.

**Missing-modality robustness.** Tab. 4 evaluates segmentation when Modalities are Not available ($MN \in \{1, 2, 3\}$) or with Full Modalities (FM). Across all three targets, Tumor Core (TC), Enhancing Tumor (ET), and Whole Tumor (WT), our method achieves the best performance in average over all settings, with 81.09% (TC), 66.27% (ET), and 90.09% (WT), outperforming the strongest baselines (e.g., PUIR: 79.78%/63.49%/87.63%). From FM to MN=3, our performance drops only -14.97 (TC), -19.05 (ET), -6.06 (WT), while methods such as mmFormer suffer substantially larger declines (e.g., ET: 79.91 → 48.86, -31.05). Meanwhile, our Std. is the smallest or among the smallest across all targets, indicating stable behavior under random missing-modality patterns.

### 5.2.3. LABEL EFFICIENCY ANALYSIS

Fig. 4 shows performance as the labeled budget increases (we randomly sample 5%, 10%, 50%, and 100% of the training set). For the segmentation task, all methods improve with more data, but TACO consistently leads across budgets, with the largest margins in the low-label regime (5–10%), indicating stronger data efficiency. The gap narrows at 100%, yet our method remains best. For classification, TACO grows steadily from 5% to 100%, while S$^2$DC plateaus and BrainMVP even degrades at high budgets. Overall, our method provides the most robust gains across data scales and tasks.

### 5.3. Ablation Study

**Ablation of losses.** The results in the Table 5 demonstrate the effectiveness of incorporating both $\mathcal{L}_{inter}$ and $\mathcal{L}_{intra}$. Introducing either $\mathcal{L}_{inter}$ or $\mathcal{L}_{intra}$ leads to performance improvements. Overall, these results highlight that the proposed approach, by leveraging both intra-instance and inter-instance topological consistency, enhances performance across both segmentation and classification tasks.

**The impact of $Top_\omega$.** We investigate the impact of $Top_\omega$ in Equation 4 by pre-training models with $\omega \in [1, 3, 5, 7]$ and evaluating on BraTS-MET and ADHD. As shown in Fig. 4(b), performance on BraTS-MET improves steadily, peaking at $Top_5$ before slightly decreasing. Performance

on ADHD peaks at $\omega = 3$ and declines thereafter. Based on these results, we select $\omega = 5$ for subsequent experiments to balance both tasks.

*Table 4.* Missing-modality segmentation on BraTS18. MN: number of missing modalities; FM: full modalities; All settings: averages over all MN cases. Mean/Std.: The average and standard deviation of different modality combinations under the current MN. Detailed results are provided in the Appendix.

| Method | All settings | | FM | MN=1 | | MN=2 | | MN=3 | |
|---|---|---|---|---|---|---|---|---|---|
| | Mean | Std. | - | Mean | Std. | mean | Std. | Mean | Std. |
| | | | | | | Tumor Core | | | |
| RFNET | 76.08 | 6.78 | 83.4 | 80.63 | 4.59 | 76.57 | 5.68 | 68.95 | 6.87 |
| mmFormer | 76.43 | 5.94 | 82.22 | 79.78 | 5.38 | 76.55 | 6.16 | 71.45 | 6.15 |
| SPA | 74.8 | 6.29 | 82.23 | 78.99 | 4.53 | 75.01 | 5.75 | 68.44 | 6.23 |
| M$^3$AE | 72.67 | 4.65 | 80.29 | 77.61 | 4.32 | 73.37 | 5.35 | 64.79 | 5.75 |
| M2F | 73.69 | 5.52 | 80.34 | 77.48 | 5.19 | 74.17 | 6.07 | 67.51 | 6.63 |
| BrainMVP | 76.23 | 6.56 | 83.83 | 78.97 | 5.95 | 73.33 | 5.24 | 68.79 | 6.33 |
| PUIR | 79.78 | 5.29 | 86.72 | 83.64 | 2.29 | 79.56 | 2.83 | **74.51** | 2.22 |
| **TACO** | **81.09** | 6.48 | **87.45** | **84.32** | 3.58 | **80.1** | 3.89 | 72.48 | 3.26 |
| | | | | | | Enhancing tumor | | | |
| RFNET | 59.31 | 13.17 | 73.65 | 66.91 | 14.54 | 59.17 | 16.57 | 48.35 | 17.72 |
| mmFormer | 62.14 | 11.67 | **79.91** | **71.54** | 17.11 | 61.77 | 17.43 | 48.86 | 16.77 |
| SPA | 58.92 | 10.87 | 73.4 | 68.44 | 12.9 | 58.05 | 14.98 | 47.1 | 14.74 |
| M3AE | 55.98 | 13.35 | 73.79 | 65.09 | 15.94 | 55.53 | 18.48 | 43.09 | 18.23 |
| M2F | 58.84 | 12.12 | 75.26 | 66.67 | 13.83 | 58.99 | 16.33 | 46.7 | 16.71 |
| BrainMVP | 61.35 | 9.36 | 73.64 | 62.25 | 10.7 | 58.23 | 17.06 | 51.29 | 7.32 |
| PUIR | 63.49 | 4.33 | 70.64 | 64.44 | 6.62 | 63.87 | 6.66 | 60.19 | 10.85 |
| **TACO** | **66.27** | 8.1 | 75.31 | 69.44 | 11.2 | **64.05** | 11.97 | 56.26 | 12.28 |
| | | | | | | Whole tumor | | | |
| RFNET | 83.92 | 5.34 | 89.27 | 87.25 | 1.39 | 84.95 | 5.3 | 77.7 | 8.16 |
| mmFormer | 84.84 | 4.48 | 88.26 | 87.59 | 1.25 | 85.36 | 4.16 | 80.45 | 6.81 |
| SPA | 84.52 | 5.04 | 89.03 | 87.81 | 2.94 | 85.26 | 5.24 | 78.98 | 6.6 |
| M3AE | 81.52 | 3.53 | 86.82 | 85.64 | 2.4 | 82.43 | 5.08 | 74.74 | 6.67 |
| M2F | 83.88 | 4.7 | 88.72 | 87.3 | 1.99 | 84.62 | 4.52 | 78.13 | 7.05 |
| BrainMVP | 84.22 | 6.48 | 89.83 | 88.78 | 0.83 | 82.43 | 7.57 | 75.82 | 10.5 |
| PUIR | 87.63 | 2.60 | 91.19 | 89.49 | 1.82 | 87.45 | 3.0 | 85.17 | 3.74 |
| **TACO** | **90.09** | 2.65 | **92.56** | **91.51** | 1.12 | **89.78** | 2.68 | **86.5** | 3.92 |

**The impact of $\delta$.** Fig. 4 (c) shows the impact of the margin $\delta$ in $L_{intra}$ and $L_{inter}$. For BraTS-MET, performance declines with increasing $\delta$, while ADHD performs best at $\delta = 0.3$. To balance both tasks, we select $\delta = 0.3$.

*Table 5.* Ablation results of $L_{intra}$ and $L_{inter}$.

| Baseline($L_{uni}$) | $L_{intra}$ | $L_{inter}$ | BraTS-MET | | | | ADHD | | |
|---|---|---|---|---|---|---|---|---|---|
| | | | ET | TC | WT | avg. | ACC | AUC | F1 |
| ● | | | 58.24 | 61.54 | 65.10 | 61.63 | 64.94 | 65.17 | 63.06 |
| ● | ● | | **64.63** | 68.50 | 69.74 | 67.62 | 64.94 | 67.91 | 57.61 |
| ● | | ● | 63.94 | 68.20 | 69.05 | 67.07 | 65.8 | 70.60 | 61.44 |
| ● | ● | ● | 64.41 | **69.52** | **70.6** | **68.17** | **67.53** | **71.14** | **63.82** |

## 5.4. Analysis of IM-NRC

**Inter-patient embedding structure (Fig. 5).** We visualize patch embeddings from four patients as shown in Fig. 5. Features from a randomly initialized model collapse into a single amorphous cloud, indicating no meaningful structure. BrainMVP forms a few large clusters dominated by a single color, i.e., features are separated mainly by patient identity, suggesting patient-specific memorization that can hurt cross-patient transfer. $S^2DC$ produces many fragmented clusters with mixed colors, while some local separability exists, but the manifold is noisy and lacks global organization. TACO yields a compact distribution, repeatedly occurring clusters where colors are well interleaved. This pattern indicates modality/instance-invariant alignment and neighborhood consistency across instances, supporting the observed gains in both segmentation and classification.

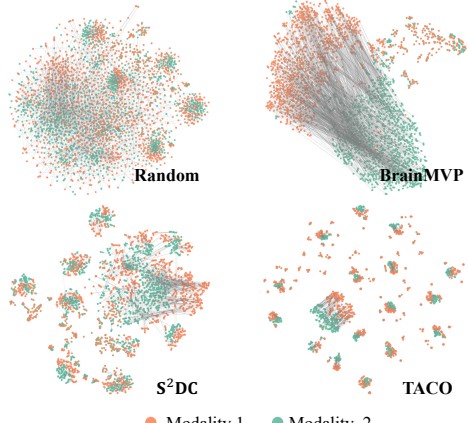

*Figure 6.* t-SNE visualization of intra-instance token feature alignment across 2 modalities. "Random" refers to a model with random initialization. TACO shows superior alignment, with more coherent clusters.

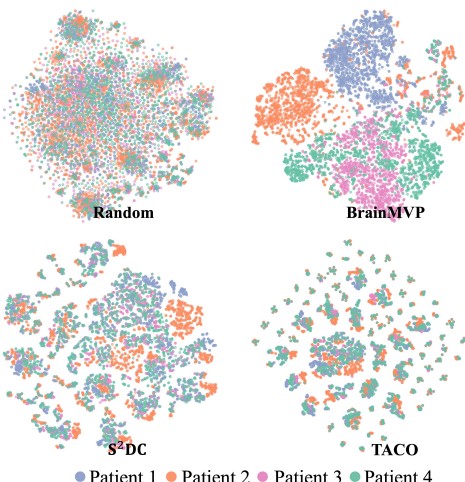

*Figure 5.* The t-SNE visualization of token-level representations across patients. TACO achieves clear and well-organized clusters across patients, demonstrating superior cross-instance learning.

**Intra-patient multi-modal embedding structure (Fig. 6).** We color points by modality as shown in Fig. 6. For patches from two modalities in the same patient, grounded patch correspondences are found, with aligned pairs highlighted by grey lines. Random embeddings collapse into a noisy cloud with no clear structure. BrainMVP shows a clear modality split (orange vs. green) with many long inter-cluster links, indicating modality-specific embeddings and hindering cross-modal transfer. $S^2DC$ forms modality-dominated islands, with frequent edge crossings that reveal unstable neighborhoods. In contrast, TACO produces compact, well-separated micro-clusters with interleaved colors and mostly intra-cluster edges, demonstrating modality-invariant alignment and stable neighborhood consistency, which supports its robustness in downstream tasks and missing-modality settings.

**Clustering on Brain Tissues (Fig. 7).** We generate brain

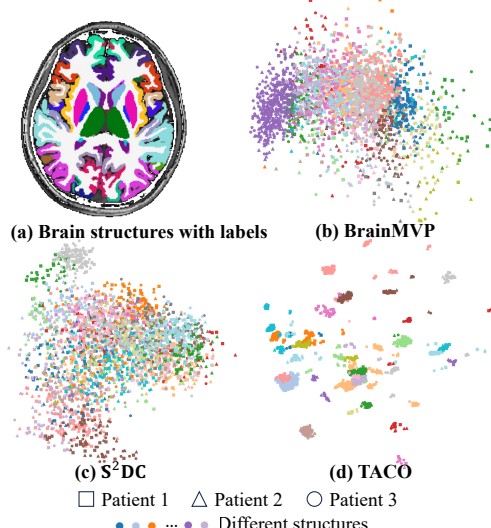

*Figure 7.* Visualization of brain structure representations across methods. (a) Ground-truth labels of brain structures, with 10 structures selected for the visualizations in (b)(c)(d) methods. TACO shows superior performance in region clustering.

structure labels for 3 MRI T1 volumes from 3 individuals in the Queensland Twin Adolescent Brain (QTAB) dataset (Strike et al., 2023) using the tool (Billot et al., 2023). Using pixel-level structure labels, Fig. 7 presents token-level visualizations of brain structures across different methods. In Fig. 7(a), the ground-truth labels show the expected anatomical segmentation. The representations from BrainMVP and $S^2DC$ exhibit some clustering of patient-specific features, but a less distinct separation between anatomical regions is observed. TACO provides the most coherent representation, with well-defined clusters corresponding to both anatomical regions and patient identities. Our method achieves the best cross-patient and cross-structure alignment, demonstrating superior generalization across individuals and structures.

## 6. Conclusion

This work introduces a novel approach to learning topological consistency in brain structures across instances, extending beyond traditional instance-level self-supervision through the proposed IM-NRC principle. Extensive evaluation on seven benchmark datasets, along with visualizations of patch features, further highlights the efficacy of our approach. **Limitation.** Future work can expand TACO to include more modalities and anatomical structures, such as whole-body PET-CT imaging, to increase its applicability.

## Acknowledgments

This work was supported by Shanghai Municipal Science, Technology Major Project (2023SHZDZX02 and 2017SHZDZX01 to L.J.), and the Industrial Alignment Funding - Pre-Positioning Programme – (IAF-PP Grant ID: H24J4a0044) awarded to Prof. Weimiao Yu and his iDMP lab. The computations in this research were performed using the CFFF platform of Fudan University.

## Impact Statement

This paper presents a method to enhance generalization ability across instances/people and modalities in medical self-supervised learning, which is crucial for foundation-modeling in medical imaging. This could potentially reduce reliance on expensive expert annotations, improve data efficiency, and enable more robust downstream performance in clinically relevant tasks such as segmentation, detection, and disease classification, particularly in settings where labeled data are scarce or heterogeneous across institutions.

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

# A. Explanation of Concepts

## A.1. Symbols and Denotations

Tab. 10 shows the symbols and the corresponding denotations.

## A.2. IM-NRC

The Instance-agnostic Cross-Modal Neighborhood Ranking Consistency (IM-NRC) principle proposes a phenomenon that consistently occurs in physiology. It is built upon the concept of topological consistency, which refers to the inherent connectivity and structure of an object. In the human body, especially the brain, physiological structures are naturally interconnected in ways that cannot be altered without fundamentally changing their organization. Moreover, this connectivity is preserved not only during rigid transformations but also under non-rigid deformations. Thus, we adopt this relational position consistency rather than relying on strictly constrained patch alignment. Fig. 8a shows this motivation.

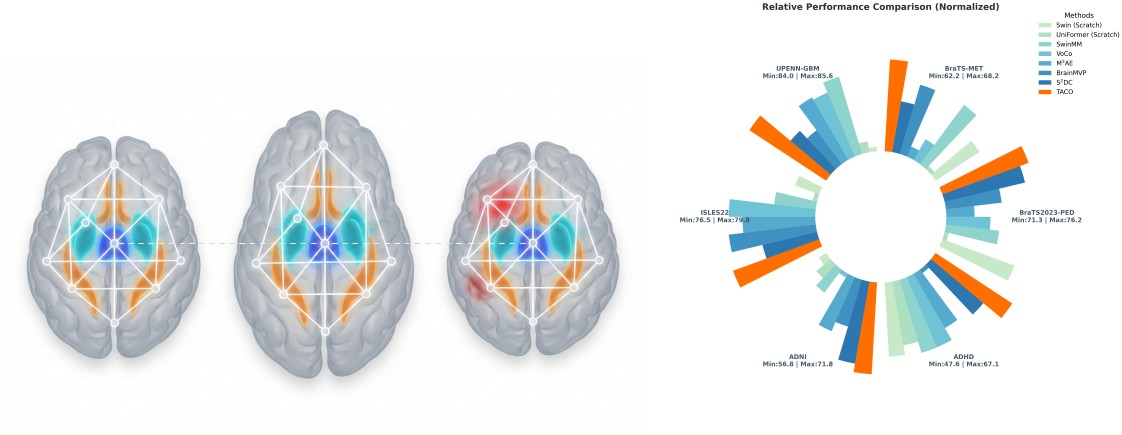

*(a)* Consistency of spatial relationships.    *(b)* Relative performance comparison.

*Figure 8.* Visualization of (a) cross-subject spatial consistency and (b) normalized relative performance metrics.

# B. Comparison with SOTA

Fig. 8b presents a relative performance comparison with min-max scaling normalization. From the figure, it is evident that TACO exhibits standout performance across all datasets.

**Compared with GVSL (He et al., 2023).** GVSL maintains the geometric consistency with predictable explicit transformation. Specifically, the model predicts the affine matrix and deformable mappings between two images. As illustrated in GVSL, the motivation and techniques aim to cluster features with the same semantics across different images, which is fundamentally different from ours. First, as indicated by the title and problem formulation, our method considers the multi-modal data rather than two augmented views derived from a single modality. The transformation between different modalities can not be explicitly presented. Second, we do not directly optimize triplet distances across two views/modalities. Instead, we define triplets in one modality and apply the corresponding ranking/metric loss in the other modality, thereby enforcing cross-modal geometric consistency of the representation space. Direct clustering of features across images/views can preserve only global organ-structure consistency within one modality or instance, while overlooking the conformal correspondences required for robust cross-modality or cross-instance alignment. These objectives and mechanisms are therefore fundamentally different from GVSL. **GVSL results are not included in our tables for the following reasons.** According to the official GVSL repository, the method is not intended to be applied to MRI data. In our own attempts to reproduce GVSL under the MRI setting, we also encountered consistent convergence issues, which prevented us from obtaining stable and reliable results for a fair comparison.

## C. Robustness to registration error

We simulate rigid registration residuals by applying random translations/rotations to one modality only, while keeping the other modality unchanged, and then re-evaluate the latent space. The performance is in the following table:

*Table 6.* Robustness under different levels of rigid errors.

| Setting | Pos. cos dist | Top-1 retrieval | MNN selected ratio |
|---|---|---|---|
| Clean | 0.0175 | 93.31% | 89.75% |
| Mild rigid error (max 2 vox / 2°) | 0.0184 | 92.98% | 88.96% |
| Moderate error (max 4 vox / 5°) | 0.0209 | 91.55% | 86.61% |
| Strong error (max 8 vox / 10°) | 0.0277 | 86.59% | 79.06% |

## D. Distance between positive and negative tokens

We provide quantitative latent-space disentanglement results of our method across 3,396 pretraining subjects. The hard negative point means the nearest negative point to the anchor point. The performance illustrates the strong separation between positive and negative samples, as shown in the following table:

*Table 7.* Distance between positive and negative token.

| Metric | Value |
|---|---|
| Pos cos dist | $0.0175 \pm 0.0004$ |
| Neg cos dist | $0.5502 \pm 0.0003$ |
| Hard-neg cos dist | $0.0616 \pm 0.0002$ |
| Neg–Pos gap | 0.5327 |
| Hard-neg–Pos gap | 0.0441 |
| Top-1 retr. acc. | $93.31\% \pm 0.26\%$ |
| Top-5 retr. acc. | $99.27\% \pm 0.06\%$ |
| Pairwise rank acc. | $99.45\% \pm 0.03\%$ |

## E. Computational Complexity

We compare the computational complexity with Swin-B-based pretrained models from the number of parameters and TFLOPS, as Tab. 8 shows. All inputs are set as $96 \times 96 \times 96$.

*Table 8.* A comparison of computational complexity

| Methods | Parameters | TFLOPS |
|---|---|---|
| VoCo | 127.44M | 15.48 |
| $S^2DC$ | 63.93M | 5.09 |
| TACO | 39.96M | 4.40 |

From the open-source code, VoCo, which uses both a teacher and a student model, includes 5 extension heads for different outputs from the Swin-B encoder. This setup introduces additional parameters and increases the TFLOPS during pre-training. Similarly, $S^2DC$ also employs a teacher-student model structure, adding extra parameters. In contrast, TACO, which only has a small decoder head and does not include teacher-student structures, demonstrates the best performance in terms of parameters and TFLOPS.

## F. Experiments on OpenMind

To ensure a rigorous evaluation, we integrated our framework into the SSL3D OpenMind ecosystem. We evaluated TACO against these SOTA baselines across two segmentation datasets, ISLES and MSLS2017. The reported evaluation metrics are DSCO(Dice Similarity Coefficient) and NSD(Normalized Surface Distance). We provide both the average and the standard deviation value. The results are provided at Table 9

*Table 9.* Comparison of pre-training methods on ISLES and MSLS2017.

| Pre-training method | ISLES | | | | MSLS2017 | | | |
|---|---|---|---|---|---|---|---|---|
| | DSC Avg. | DSC Std. | NSD Avg. | NSD Std. | DSC Avg. | DSC Std. | NSD Avg. | NSD Std. |
| MAE (best in OpenMind) | 74.02 | 2.38 | 71.82 | 0.85 | 83.02 | 4.46 | 92.66 | 4.82 |
| TACO (Ours) | 75.34 | 2.44 | 72.12 | 1.28 | 83.96 | 4.50 | 93.50 | 4.81 |

# G. Implementation Details

All code, including pre-training, fine-tuning, and data splits, will be made publicly available upon publication.

*Table 10.* Summary of notations used throughout the paper.

| Symbol | Description |
|---|---|
| $i, j, t$ | Modality categories |
| $a, p, n, k$ | Patch indices |
| $h, g$ | Individual IDs |
| $x_i^h$ | Image data from individual $h$ in modality $i$ |
| $\mathcal{E}$ | Transformer encoder |
| $\mathcal{D}$ | Decoder |
| $Z_i^h$ | Output tokens from $x_i^h$ through $\mathcal{E}$ |
| $K$ | Output token number from $x_i^h$ through $\mathcal{E}$ |
| $z_{i,k}^h$ | The $k$-th token in $Z_i^h$ |
| $C_{i \to j}^{h \to g}$ | Correspondence from modality $i$ to $j$ and from individual $h$ to $g$ |
| $\tilde{C}_{i \to j}^{h}$ | Correspondence from modality $i$ to $j$ within individual $h$ |
| $\mathcal{P}_{i,h}^\omega(a)$ | The neighborhood set of $\omega$ nearest patches' indices in $Z_i^h$ at $a$-th patch. |
| $\mathcal{N}_{i,h}(a)$ | The non-neighborhood set of with respect to $\mathcal{P}_{i,h}^\omega(a)$. |
| $\mathcal{P}_{i,h}'^\omega(a), \mathcal{P}_{i \to t, h \to g}'^\omega(a)$ | The set of corresponding patch indices with respect to the neighborhood set. |
| $\mathcal{N}_{i,h}'^\omega(a), \mathcal{N}_{i \to t, h \to g}'^\omega(a)$ | The set of corresponding patch indices with respect to the non-neighborhood set. |
| $\text{Tri}_{i \to j, a, h}^{\text{intra}}, \text{Tri}_{i \to t, a, h \to g}^{\text{inter}}$ | The set of triplet patches' indices at $a$-th patch. |
| $V_{i,t}^{h,g}$ | The set of solid indices pairs for anchor points selection. |
| $\mathcal{L}_{\text{uni}}, \mathcal{L}_{\text{intra}}, \mathcal{L}_{\text{inter}}$ | Loss functions used in our method. |

## G.1. Pre-training

For all reproduced open-source methods, we use their official implementations. For TACO, we utilize the Swin-B model for pretraining, paired with a 6-layer CNN-UPSAMPLE-InstanceNorm3d decoder. The hyperparameters are set as follows: learning rate=$3e-4$, weight decay=$1e-5$, iteration=10k, optimizer=AdamW, and input volum size=$96 \times 96 \times 96$. Additionally, we apply a cosine annealing scheduler with warmup.

## G.2. Fine-tuning

All methods, including those trained from scratch, are evaluated under consistent fine-tuning settings. For multi-modal tasks, we follow the same strategy as BrainMVP, initializing the encoder with pre-trained weights, except for the patch embedding.

**Segmentation tasks.** Swin-B-based pre-trained models follow the Swin UNETR structure, ViT-B-based pre-trained models follow the UNETR structure, and UniFormer-based pre-trained models follow the UniUNET structure (as in BrainMVP). For M$^3$AE, the UNET3D structure is used. The hyperparameters are set as follows: learning rate=$1e-4$, weight decay=$1e-5$, epochs=300, optimizer=AdamW, and input volume size=$96 \times 96 \times 96$. Except for ISLES, we adopt input volume size=$128 \times 128 \times 128$. During inference, a sliding window size of 0.75 and the same input volume size are used. A cosine annealing scheduler with warmup is also applied. The data is split as 6:1:3 for training, validation, and testing, respectively, with the model yielding the best validation result being selected for evaluation on the test dataset.

**Classification tasks.** For those tasks, all pre-trained models adopt their encoder (i.e., Swin-B, ViT-B, UniFormer, and 3D CNN, respectively) for classification. The hyperparameters are set as follows: learning rate=$1e-4$, weight decay=$1e-5$,

*Table 11.* The results of missing modalities. ● represents the missed modality.

| Method | MN=3 | | | | MN=2 | | | | | | MN=1 | | | | FM |
|---|---|---|---|---|---|---|---|---|---|---|---|---|---|---|---|
| FLAIR | | ● | ● | ● | | | ● | ● | | ● | | | | ● | |
| T1 | ● | | ● | ● | | | | | ● | ● | | | ● | | |
| T1ce | ● | ● | | ● | | ● | | ● | ● | | | ● | | | |
| T2 | ● | ● | ● | | ● | ● | ● | | | | ● | | | | |
| *Tumor Core* | | | | | | | | | | | | | | | |
| RFNET | 64.03 | 74.53 | 58.63 | 61.95 | 79.2 | 77.45 | 69.25 | 67.48 | 67.98 | 78.85 | 80.15 | 70.75 | 79.4 | 80.15 | 80.29 |
| mmFormer | 67.8 | 77.32 | 64.56 | 64.08 | 81.51 | 79.43 | 69.14 | 70.63 | 68.6 | 80.75 | 81.75 | 70.92 | 81.74 | 81.55 | 82.23 |
| SPA | 65.86 | 65.27 | 78.26 | 66.4 | 72.99 | 83.23 | 70.66 | 81.25 | 70.66 | 80.63 | 83.22 | 73.89 | 83.36 | 82.05 | 83.4 |
| M$^3$AE | 69.4 | 65.45 | 79.12 | 71.84 | 79.9 | 70.45 | 82.79 | 81.17 | 71.62 | 73.35 | 81.78 | 82.42 | 73.31 | 81.61 | 82.22 |
| M2F | 65.79 | 63.29 | 77.31 | 63.64 | 70.38 | 79.93 | 68.01 | 79.62 | 67.68 | 79.37 | 80.65 | 69.73 | 80.01 | 79.53 | 80.34 |
| PUIR | 75.83 | 71.2 | 75.29 | 75.71 | 80.66 | 83.6 | 79.23 | 74.83 | 79.51 | 79.52 | 83.92 | 82.78 | 86.65 | 81.22 | 86.72 |
| TACO | 71.65 | 75.07 | 68.22 | 74.99 | 86.09 | 78.14 | 83.64 | 77.12 | 79.19 | 76.4 | 86.62 | 87.36 | 79.45 | 83.88 | 87.45 |
| *Enhancing tumor* | | | | | | | | | | | | | | | |
| RFNET | 38.69 | 69.22 | 30.89 | 33.56 | 71.4 | 70.9 | 38.53 | 41.91 | 40.9 | 69.51 | 71.61 | 43.37 | 71.17 | 74.2 | 73.79 |
| mmFormer | 40.08 | 72.19 | 38.89 | 37.23 | 73.11 | 73.06 | 40.64 | 42.27 | 43.65 | 75.56 | 43.34 | 81.74 | 73.36 | 75.31 | 73.4 |
| SPA | 39.85 | 41.39 | 70.43 | 41.72 | 45.99 | 73.07 | 45.25 | 72.87 | 45.25 | 72.59 | 73.52 | 47.56 | 73.01 | 73.55 | 73.65 |
| M$^3$AE | 37 | 38.41 | 75.8 | 44.22 | 78.09 | 45.2 | 79.36 | 78.16 | 41.71 | 48.12 | 79.14 | 80.06 | 47.63 | 79.31 | 79.91 |
| M2F | 37.99 | 37.79 | 71.74 | 39.28 | 43.37 | 74.66 | 45.42 | 73.48 | 43.5 | 73.48 | 73.56 | 45.93 | 73.15 | 74.03 | 75.26 |
| PUIR | 67.45 | 54.83 | 70.86 | 47.63 | 69.38 | 52.91 | 70.1 | 59.45 | 67.44 | 63.91 | 70.79 | 57.78 | 69.42 | 59.76 | 70.64 |
| TACO | 47.16 | 74.34 | 50.61 | 52.96 | 74.74 | 51.76 | 74.44 | 75.66 | 53.35 | 54.39 | 75.34 | 74.75 | 52.65 | 75.02 | 75.31 |
| *Whole tumor* | | | | | | | | | | | | | | | |
| RFNET | 80.52 | 67.06 | 68.42 | 82.96 | 82.57 | 71.97 | 85.82 | 83.25 | 86 | 84.94 | 86.06 | 86.53 | 84.34 | 83.61 | 86.82 |
| mmFormer | 84.09 | 72.85 | 73.37 | 85.6 | 85.97 | 76.93 | 87.09 | 86.09 | 87.55 | 87.94 | 88.36 | 88.16 | 88.74 | 85.96 | 89.03 |
| SPA | 85.77 | 72.69 | 71.95 | 80.4 | 87.82 | 87.97 | 88.27 | 75.57 | 88.27 | 81.8 | 88.3 | 88.78 | 89.06 | 82.87 | 89.27 |
| M3AE | 87.78 | 74.69 | 74.91 | 84.43 | 76.09 | 84.48 | 89.63 | 84.4 | 88.64 | 88.91 | 84.04 | 89.29 | 88.58 | 88.45 | 88.26 |
| M2F | 85.72 | 72.48 | 71.78 | 82.53 | 87.73 | 87.66 | 84.35 | 76.03 | 87.69 | 84.27 | 88.17 | 88.22 | 88.47 | 84.32 | 88.72 |
| PUIR | 89.23 | 81.73 | 82.26 | 87.45 | 89.74 | 89.03 | 88 | 81.92 | 89.72 | 86.27 | 89.12 | 90.5 | 91.25 | 87.1 | 91.19 |
| TACO | 91.23 | 83.4 | 83.15 | 88.22 | 91.51 | 91.69 | 89.78 | 84.86 | 91.87 | 88.99 | 91.79 | 92.54 | 91.79 | 89.92 | 92.56 |

epochs=50, optimizer=Adam, and input volume size=$128 \times 128 \times 128$. A cosine annealing scheduler is applied. Same as segmentation, the datasets are split as 6:1:3 for training, validation, and testing, respectively, with the model yielding the best validation result being selected for evaluation on the test dataset.

### G.3. Pre-training Datasets

**BraTS2021** Multimodal Brain Tumor Segmentation Challenge focuses on glioma segmentation. BraTS2021 includes T1, T1CE, T2, and FLAIR modalities with 1470 patients.

**BraTS2023-SSA** is glioma segmentation in sub-saharan Africa patient population (brats-africa), including T1, T1CE, T2, and FLAIR modalities with 75 patients.

**BraTS2023-MEN** is a meningioma segmentation challenge with 1141 patients.

**UCSF-PDGM** UCSF-PDGM is a publicly released preoperative MRI dataset of diffuse glioma, jointly developed by the University of California San Francisco (UCSF) and the University of Pennsylvania (UPenn). It includes 501 patients with T1, T1CE, T2, FLAIR, DWI, and ADC modalities.

**IXI** Our MRI dataset of IXI is extracted from Information eXtraction from Images. To keep the same as BrainMVP, we also select 568 patients. It has T1, T2, MRA, and PD modalities.

### G.4. Downstream Datasets

**BraTS2023-PED.** The BraTS2023-PEDs dataset is a multi-institutional, retrospective collection of pediatric brain tumor MRI scans, assembled for the 2023 edition of the BraTS 2023 Challenge with 99 patients. It can be seen as an out-of-distribution dataset for pretraining models (pediatric vs. brain), which also has T1, T1CE, T2, and FLAIR modalities. The original annotations are Non-Enhancing Tumor, Cystic Component, Edema, and Enhancing Tumor. For the final annotations for model training, the dataset has 3 labels: Nonenhancing Component (NC) (Label 1), Edema (ED) (Label 2),

and Enhancing Tumor (ET) (Label 3).

**BraTS2023-MET.** The BraTS2023-MET dataset, with 238 patients and T1, T1CE, T2, and FLAIR modalities, focuses on the segmentation of brain metastases on pre-treatment multiparametric MRI. The original annotations are: Necrosis, Edema, and Enhancing.

**UPENN-GBM.** The dataset includes MRI data from 630 patients diagnosed with de novo glioblastoma (adult, initial diagnosis). In this paper, we follow BrainMVP, adopting 127 patients. The original annotations are: Edema/Invasion, Enhancing Tumor, Necrosis.

**ISLES22.** ISLES (Ischemic Stroke Lesion Segmentation) is a recurring challenge for automatic segmentation of stroke lesions in MRI, with 238 patients. It has DWI, ADC, and FLAIR modalities. There is only one label (Stroke Lesion).

**BraTS2018.** BraTS2018 (Brain Tumor Segmentation Challenge 2018) dataset has expert manual annotations for sub-regions: Enhancing tumour (ET), Necrotic/non-enhancing tumour core (NCR/NET), Peritumoral edema (ED). These sub-regions can be grouped into 'Whole Tumor' (WT = ET + NCR/NET + ED), 'Tumour Core' (TC = ET + NCR/NET), and 'Enhancing Tumour' (ET) for evaluation. In this work, we select 1470 patients for the missing modality task.

**ADNI.** The ADNI is a large multicenter study launched in 2003 in the United States and Canada. In this paper, the dataset is classified as NC (Normal Control) and AD (Alzheimer's disease), with 697 patients, and the T1 modality.

**ADHD-200.** ADHD-200 is a large multi-site open-access dataset compiled to support research on ADHD and related neuroimaging biomarkers. In this work, we adopt 767 patients with T1 modality. The dataset is classified as ADHD and NC.

**The details of the pretraining and downstream datasets can be viewed in Table 2.**

### G.5. Details of Missing Modality

For the missing-modality task, we simplify the approach by using a random missing-modality training strategy. In this setup, 1 to 3 out of the 4 modalities will be randomly omitted. Our training framework utilizes the official BraTS21 implementation in Monai Swin UNETR. The hyperparameters are set as follows: learning rate=$1e-3$, batch size=1, epoch=300, optimizer=AdamW, and input volum size=$96 \times 96 \times 96$. During inference, a sliding window size of 0.75 and the same input volume size are used. A cosine annealing scheduler with warmup is also applied. The data split follows the official split in Monai Swin UNETR.

Tab. 11 shows the results of all missing-modality settings.

## H. Pseudo Code of $L_{intra}$ and $L_{inter}$

Algorithm 1 and 2 shows the pseudo code of $L_{intra}$ and $L_{inter}$.

---

**Algorithm 1** Intra-Instance Constraint $\mathcal{L}_{intra}$.

---

1: **Input:**
2:    $Z_i^h$ (output tokens from $x_i^h$)
3:    $Z_j^h$ (output tokens with shape $K$ from $x_j^h$)
4:    $C_{i \rightarrow j}^h$ (Correspondence from modality $i$ to $j$,
5:      defaulting to an equivalent patch-index mapping.)
6:    $\omega$ (number of neighbors, default 5)
7:    $\delta$ (margin, default 0.1)
8:    $d(\cdot, \cdot)$ (distance metric, default 'cosine')
9: **Calculate $\mathcal{L}_{\text{intra}}$:**
10:    $L_{\text{intra}} = 0$
11:    **for** $h$ in $H$:{Each instance in the batch}
12:       $D_h \leftarrow d(Z_i^h, Z_i^h)$ {Compute pairwise distance between all tokens of $Z_i^h$}
13:       $\mathcal{P}_{i,h}^\omega \leftarrow D_h.argsort()[:, 1 : k+1]$ {Find the $Top_\omega$ nearest neighbors for each sample}
14:       **for** $a$ in arange(K):
15:       $a' \leftarrow C_{i \rightarrow j}^h(a)$
16:          **for** $p$ in $\mathcal{P}_{i,h}^\omega(a)$:
17:            $p' \leftarrow C_{i \rightarrow j}^h(p)$
18:            $n \leftarrow \text{random\_select}(D_h.argsort()[:, k+2 :])$
19:            $n' \leftarrow C_{i \rightarrow j}^h(n)$
20:            $d_{pos} \leftarrow d(Z_j^h[a'], Z_j^h[p'])${Positive}
21:            $d_{neg} \leftarrow d(Z_j^h[a'], Z_j^h[n'])${Negative}
22:            $\mathcal{L}_{\text{intra}} \leftarrow \mathcal{L}_{\text{intra}} + \text{ReLU}(d_{pos} - d_{neg} + \delta)$
23:          **end for**
24:       **end for**
25:    **end for**
26:    $\mathcal{L}_{\text{intra}} \leftarrow \text{mean}(\mathcal{L}_{\text{intra}})$
27: **return** $\mathcal{L}_{\text{intra}}$

---

---

**Algorithm 2** Inter-instance Constraint $\mathcal{L}_{inter}$.

---

1: **Input:**
2:    $Z_i^h$ (output tokens from $x_i^h$)
3:    $Z_j^g$ (output tokens from $x_j^g$)
4:    $\omega$ (number of neighbors, default 5)
5:    $\delta$ (margin, default 0.1)
6:    $d(\cdot, \cdot)$ (distance metric, default 'cosine')
7: **Calculate $\mathcal{L}_{\text{inter}}$:**
8:    **for** $(h, g)$ in $H$:{Paired instance in the batch}
9:       $D \leftarrow d(Z_i^h, Z_j^g)$ {Distance matrix between $Z_i^h$ and $Z_j^g$}
10:      $D_{min1} \leftarrow D.\text{argmin}(\text{dim} = 1)$
11:      $D_{min0} \leftarrow D.\text{argmin}(\text{dim} = 0)$
12:      $\mathcal{M} \leftarrow \text{Arange}(K) == D_{min0}[D_{min1}]$
13:      $V_{i,j}^{h,g} \leftarrow \text{Arange}(K)[\mathcal{M}] \times D_{min1}[\mathcal{M}]$ {mutual nearest neighborhood}
14:      $D_{h,i} \leftarrow d(Z_i^h, Z_i^h)$
15:      $\mathcal{P}_{i \to j, h \to g}^{\omega} \leftarrow D_{h,i}.argsort()[:, 1 : k + 1]$ {Find the $Top_{\omega}$ nearest neighbors for each sample}
16:      **for** $(a, a')$ in $V_{i,j}^{h,g}$:
17:        **for** $p$ in $\mathcal{P}_{i \to j, h \to g}^{\omega}(a)$:
18:          $p' \leftarrow \text{find } (p, p') \text{ in } V_{i,j}^{h,g}$
19:          $n \leftarrow \text{random\_select}(D_{h,i}.argsort()[:, k + 2 :])$
20:          $n' \leftarrow \text{find } (n, n') \text{ in } V_{i,j}^{h,g}$
21:          $d_{pos} \leftarrow d(Z_j^g[a'], Z_j^g[p'])${Positive}
22:          $d_{neg} \leftarrow d(Z_j^g[a'], Z_j^g[n'])${Negative}
23:          $\mathcal{L}_{\text{inter}} \leftarrow \mathcal{L}_{\text{inter}} + \text{ReLU}(d_{pos} - d_{neg} + \delta)$
24:        **end for**
25:      **end for**
26:    **end for**
27:    $\mathcal{L}_{\text{inter}} \leftarrow \text{mean}(\mathcal{L}_{\text{inter}})$
28: **return** $\mathcal{L}_{\text{intra}}$

---

