# OpenReview forum: "Beyond Instance-Level Self-Supervision in 3D Multi-Modal Medical Imaging"
_ICML.cc/2026/Conference — ICML 2026 regular_

### Official Review · Reviewer_SRWu · 2026-03-06

**Soundness:** 4
**Presentation:** 3
**Significance:** 4
**Originality:** 3
**Overall Recommendation:** 5
**Confidence:** 4

**Summary:**

The paper presents a self-supervised pretraining strategy that leverages anatomic consistency to improve the learned representation space. The proposed method, TACO, introduces two training objectives: $L_{intra}$, a triplet loss that preserves local neighborhoods between different (aligned) modalities of the same patient, and $L_{inter}$, which first finds correspondences between patches from different patients and modalities and then performs a similar triplet loss. This strategy results in higher performance in multiple segmentation and classification tasks and in segmentation with missing modalities. In addition, the representation visualizations shows that the features do not cluster by patients or modalities. Instead, the clusters appear to be associated with distinct brain structures.

**Compliance With Llm Reviewing Policy:**

Affirmed.

**Final Justification:**

I thank the authors for the response to my questions. Given that I had few concerns and they were properly addressed, I will maintain my original acceptance score.

**Key Questions For Authors:**

- Considering that during fine-tuning models suffer from catastrophic forgetting, would it help to include some extra constraints to better preserve the characteristics of the feature spaced obtained with TACO?
- Given the difference in behavior of the losses shown in Fig 4.c with respect to the margin ($\delta$), would there be an advantage in selecting different margins for the $L_{inter}$ and $L_{intra}$ losses?

**Limitations:**

Yes

**Strengths And Weaknesses:**

**Strengths**
- The paper presents an SSL approach to preserve topological consistency of structures in the brain across patients and imaging modalities. This idea is highly relevant in the medical community, since the anatomy is largely consistent across subjects, but that prior is rarely explicitly leveraged.
- The method is clearly described and appears technically sound. The two proposed losses capture complementary forms of alignment: across modalities from the same patient ($L_{intra}$) and across both subjects and modalities ($L_{inter}$)
- The experimental setup and implementation details are presented clearly and appear sufficiently reproducible.
- The results show that the proposed pretraining obtains superior performance compared to training methods from scratch and starting from other state-of-the-art pretraining strategies across multiple downstream tasks. The method also demonstrates robustness to missing imaging modalities.
- The visualizations of token representations demonstrate the limitations of existing SSL pretraining methods, which tend to produce features that cluster by patients or modalities. This observation is particularly interesting since an optimal representation of anatomical structures should be invariant to those factors. The proposed pretraining approach produces clusters that better reflect this desired behavior.

**Weaknesses**
- There are some minor issues related to presentation and organization. In Section 5.1 (Downstream datasets), the text references Table 2 from the Appendix, but that table is in the main paper. In addition, Table 2 is referenced before Table 1, which could be reorganized for clarity. The results tables highlight the best method in red, which conventionally indicates bad results and may be confusing. Finally, in figure 4 there is a vertical line separating subfigures a) and b), but not between b) and c) which makes it look as if those last two also corresponded to the same experiment
- The analysis of the ablation experiment on the number of neighbors ($Top_ω$) does not reflect what is shown in figure 4.b. The text states that “Performance on ADHD peaks at ω= 3 and declines thereafter” but the figure shows that the ACC is highest at 1, lowest at 3 and after it starts slowly increasing again
- In equation 5, should it be $(p’,n’)$ instead of $(a’,n’)$ in the second line?

---

> ### Author Rebuttal · Authors · 2026-03-31
>
> We appreciate the reviewer's positive comments. Below, we provide our response.
>
> **W1** Thank you for pointing out these presentation issues. We agree that these aspects can be improved for better clarity and consistency.
>
> First, regarding the reference to Table 2 in Section 5.1, we apologize for the confusion caused by the incorrect table location description. We will revise the text to ensure that the table references are accurate and consistent with the final manuscript structure.
>
> Second, we agree that citing Table 2 before Table 1 is not ideal from an organizational perspective. In the revised version, we will reorder the discussion and table references to follow a clearer and more natural presentation flow. Regarding the table formatting, we will change the highlighting of the best results from red text to bold to avoid any potential confusion, following standard academic conventions.
>
> Third, thank you for noting the visual ambiguity in Fig. 4. We will add a vertical separator between (b) and (c\) to make the grouping of subfigures clearer.
>
> These issues are purely related to presentation and do not affect the experimental results or conclusions, but we appreciate the reviewer’s careful reading and will address them in the revision.
>
> **W2** Thank you for pointing this out: The ADHD performance shows a dip at $\omega=3$. As shown in the figure, the performance gap between the classification dataset ADHD and the segmentation dataset BraTS-MET becomes smaller when $\omega > 3$. In particular, on ADHD, the model performs worst at $\omega = 3$ and then shows a slight recovery as $\omega$ increases further. In contrast, performance on BraTS-MET reaches its peak at $\omega = 5$. Considering both the trade-off between classification and segmentation performance and the overall results across tasks, we therefore choose $\omega = 5$ in our experiments. We will revise the discussion in our paper. We thank the reviewer for their careful reading and will revise the discussion in the paper.
>
> **W3** Thanks for pointing out the typo in our equation. We will revise it in our paper.
>
> **Q1** It is true that there are catastrophic forgetting problems when the pre-trained model is finetuned on a different dataset. This topic has been widely discussed in the area of continual/incremental learning. Considering the method related to geometry in the latent space, we believe some representation-based methods [1], such as saving prototypes, may help preserve the characteristics of the feature space. However, since our paper focuses on the SSL method instead of downstream tasks, we follow standard evaluation protocols used in prior work for the downstream experiments.
>
> **Q2** Since $L_{\text{inter}}$ and $L_{\text{intra}}$ capture different geometric relationships, they may benefit from different margin values in principle. In the current paper, we use a shared margin $\delta$ mainly for simplicity and robustness: it keeps the formulation compact, reduces the hyperparameter search space, and avoids making the overall performance dependent on additional loss-specific tuning. Empirically, even with a shared margin, the method already achieves stable improvements. we agree that using separate margins for $L_{\text{inter}}$ and $L_{\text{intra}}$ could provide additional flexibility. We will clarify this point in the revised version and mention that adopting distinct margins for the two terms is a meaningful direction for future improvement.
>
>
> [1] Wang, Liyuan, et al. "A comprehensive survey of continual learning: Theory, method and application." IEEE transactions on pattern analysis and machine intelligence 46.8 (2024): 5362-5383.

---

> > ### Author Rebuttal · Reviewer_SRWu · 2026-04-02
> >
> > I thank the authors for the response to my questions. Given that I had few concerns and they were properly addressed, I will maintain my original acceptance score.

---

> > > ### Author Response · Authors · 2026-04-08
> > >
> > > We appreciate the reviewer's constructive comments. We have revised our manuscript according to the suggestions.

---

### Official Review · Reviewer_EeFS · 2026-03-07

**Soundness:** 2
**Presentation:** 3
**Significance:** 2
**Originality:** 3
**Overall Recommendation:** 4
**Confidence:** 4

**Summary:**

The paper proposes a self-supervised pre-training method specifically designed for multimodal medical images data. The underlying idea is that anatomical structures are similar across modalities and individuals. To achieve this, the authors use a reconstruction task to which they add constraints based on contrastive learning in the latent space: one intra-patient and one inter-patient. For the intra-patient constraint, the authors want the tokens (from a transformer) to have the same neighborhood from one modality to another for the same patient. The correspondence between the tokens of the two modalities is obtained by the same position in the images. For the inter-patient constraint, it is not possible to match the tokens of two patients using their spatial position. The authors therefore define the correspondence between two tokens using the mutual nearest neighborhood. Once the correspondence has been found, the same constraint is applied.

The authors evaluate themselves on seven brain MRI datasets for segmentation and classification tasks. They also evaluate the robustness of their method to a missing modality for segmentation.

**Compliance With Llm Reviewing Policy:**

Affirmed.

**Final Justification:**

The authors have responded to almost all the comments. The only remaining limitation is the limited improvement in performance compared with the best state-of-the-art methods (with no evidence of a statistically significant difference). Nevertheless, I believe the proposed method may be of interest to the community and I have improved my score from Weak reject to Weak accept.

**Key Questions For Authors:**

Weaknesses 3, 4, 8, 9

**Limitations:**

The method relies on a good registration which is not discussed.

**Strengths And Weaknesses:**

Strengths:

- Using prior knowledge about anatomical structures for self-supervised learning is relevant, and the proposed method does not require additional annotation costs (no need for an atlas, for example) or computational costs (just two additional losses for pre-training).
- The method is evaluated on numerous datasets and several tasks.
- Performance is slightly superior to state-of-the-art methods
- The paper is well written and easy to follow.

Weaknesses:
- 1/ The assumption that tokens must have similar neighborhood between different modalities does not always seem correct to me. For example, a pathology may only be visible in one modality. In this case, in the modality without visible pathology, we would expect all ‘healthy’ patches to be close together in latent space, whereas in the other modality, the corresponding tokens may correspond to “healthy” or ‘pathological’ patches that we would like to see separated in the latent space.
- 2/ By using the top-$\omega$ nearest neighbors to select the positive samples of the contrastive learning, the authors do not take into account the distance. As the negative samples are all the other samples, it could be an issue: negative and positive samples could be very closed. Would not it be better to set a margin?
- 3/ The method to select the neighbors for the inter-patient constraint is very strict. I wonder how many samples are selected using this criterion.
- 4/ The results are quite similar to state-of-the-art methods and there is no statistical significance test (in most cases, there is not even a standard deviation). I wonder whether the method is significantly better or not.
- 5/ Adding a more difficult segmentation task (segmentation of multiple sclerosis lesions, for example) might be necessary to see a difference between the methods. Increasing the Dice score by one point does not add anything for clinicians. Other metrics could also be used, such as the Hausdorff distance.
- 6/ It would have been interesting to add the best methods (S²DC and BrainMVP) in the  evaluation of robustness to missing data (Table 4).
- 7/ The authors do not evaluate their method to registration error, which is frequent and should impact the proposed method.
- 8/ The results are not the same in Table 3 and Figure 4 (BrainMVP has the best accuracy in the table and the worst in the figure). How much data was used for Table 3?
- 9/ Adding quantitative results on the latent space disentanglement would be a plus (distance between positive and negative tokens for example)
- 10/ In Figure 4, it would be interesting to add a curve without self-supervision
- 11/ In Figure 7: the shape are not visible.
- 12/ In Tables 1, 4 and 8, please add that the metric is the Dice.

Typos:
- In assumption 3.1: $z_{i,k}^h$ and not $z_{i,1}^h$

---

> ### Author Rebuttal · Authors · 2026-03-31
>
> Thanks for the comments. We provide our response below.
>
> **W1** We agree that pathology saliency varies, but our method aligns anatomical manifolds rather than pixel-level intensity: (1) Structural Invariance: Even if a lesion is subtle in one modality, it remains embedded in a consistent anatomical context and spatial organization. Our model captures these invariant structural relationships. (2) Goal of Pre-training: Our objective is to learn transferable structural representations, not fine-grained lesion equivalence. Prioritizing shared anatomy over modality-specific pathology ensures a more robust latent space for downstream tasks.
>
> **W2** Our framework focuses on cross-modal relational transfer rather than simple instance discrimination: (1) Structural Alignment: Our triplets in modality A are guided by the similarity structures of modality B. The goal is to enforce distributional consistency between modalities by transferring their relative relational structures. As long as the relative rankings of "positive" and "negative" remain consistent across modalities, structural alignment is effectively achieved, as evidenced by our visualizations. (2) Adaptivity vs. Fixed Margin: the predefined margin may be difficult to calibrate robustly because latent space density fluctuates dynamically during training and across different anatomical regions. Our Top-$w$ approach provides a more adaptive supervision signal, ensuring stable manifold alignment regardless of local density shifts.
>
> **W3** To quantify how many tokens are actually retained in the inter-instance case, we evaluate the MNN selection. Across 3,396 pretraining samples, MNN selects 14.02 of 27 candidate tokens on average (51.91%). Among the selected tokens, 94.19% remain active in the subsequent neighborhood-ranking consistency stage, corresponding to 13.20 active anchors per ordered pair. Table: [link](https://imgur.com/a/QmcL4BW).
>
> **W4, W5** Our method consistently gains average improvement across all benchmark datasets. Specifically, in segmentation, TACO improves the dataset-level average on all four datasets. It also achieves the best overall average score, improving from 76.36 to 77.47 (+1.11) over the strongest prior method. In classification, TACO also improves the dataset-level AUC/F1 on both datasets. Moreover, it achieves the best average ACC/AUC/F1 across the two classification benchmarks, improving from 81.30/82.77/77.93 to 81.86/84.12/78.70 (+0.56/+1.35/+0.77). Note that all experiments in our paper, including the implementation of other methods, are conducted with the same seed under the same evaluation protocol for fair comparison. We believe these consistent improvements across datasets, tasks, and metrics provide supportive empirical evidence that the advantage of our method is systematic rather than incidental. We also conduct experiments on the multiple sclerosis lesion dataset, MSLS2017. Following the OpenMind benchmark, we use the more robust metric, NSD, instead of HD, since HD can be dominated by outliers. The results are provided in our response to **Reviewer rbZy**. The std values are provided in the results.
>
> **W6** The results of BrainMVP in the missing-modality setting: [link](https://imgur.com/a/WmTEP8k). As the table shows, BrainMVP suffers a performance drop, highlighting the stronger modality-alignment capability of our model.
>
> **W7** We simulate rigid registration residuals by applying random translations/rotations to one modality only, while keeping the other modality unchanged, and then re-evaluate the latent space. The performance is in the following table:
> ||Pos. cos dist|Top-1 retrieval|MNN selected ratio|
> |-|-|-|-|
> |Clean|0.0175|93.31%|89.75%|
> |Mild rigid error(max 2 vox/2°)|0.0184|92.98%|88.96%|
> |Moderate error(max 4 vox/5°)|0.0209|91.55%|86.61%|
> |Strong error(max 8 vox/10°)|0.0277|86.59%|79.06%|
>
> Registration error affects the results, but performance degrades gradually under rigid perturbations rather than collapsing abruptly. Curve: [link](https://imgur.com/a/qyoh17r).
>
> **W8,W10** We corrected the BrainMVP plotting error in Figure 4 at [link](https://imgur.com/a/QpFBSUY). We also clarify that Table 3 uses the full dataset (100%).  Finally, a full non-SSL baseline is infeasible in rebuttal due to limited computational resources, but we will include it in the final version.
>
> **W9** We provide quantitative latent-space disentanglement results of our method across 3,396 pretraining subjects. The hard negative point means the nearest negative point to the anchor point. The performance illustrates the strong separation between positive and negative samples. Histogram: [link](https://imgur.com/a/8bwElKN).
> |Metric|Value|
> |-|-|
> |Pos cos dist|0.0175±0.0004|
> |Neg cos dist|0.5502±0.0003|
> |Hard-neg cos dist|0.0616±0.0002|
> |Neg−Pos gap|0.5327|
> |Hard-neg−Pos gap|0.0441|
> |Top-1 retr. acc.|93.31%±0.26%|
> |Top-5 retr. acc.| 99.27%±0.06%|
> |Pairwise rank acc.|99.45%±0.03%|
>
> **W11,W12,typo** We will revise them in our paper.

---

> > ### Author Rebuttal · Reviewer_EeFS · 2026-04-02
> >
> > Thank you for the detailed response and the additional experiments. The authors' answers are very interesting, and I think the issues raised and the answers provided should be included in the final paper. The only remaining limitation is the limited improvement in performance compared with the best state-of-the-art methods (with no evidence of a statistically significant difference). Nevertheless, I believe the proposed method may be of interest to the community.

---

> > > ### Author Response · Authors · 2026-04-08
> > >
> > > We sincerely thank the reviewer for the positive comment that our work may be of interest to the community. To address the remaining concerns regarding performance margins and statistical significance, we further extended our evaluation to an extremely challenging task: Aneurysm Segmentation using the ADAM [1] dataset.
> > >
> > > - Evaluation on Extreme Sparsity: In the ADAM dataset, the aneurysm regions occupy only ~0.0011% of the total volume.
> > > - Performance and Stability: We compared our TACO (pre-trained on 114k OpenMind volumes) against the SOTA baseline (MAE checkpoint released by the OpenMind benchmark) and a from-scratch baseline. Notably, the MAE baseline failed to converge (loss became NaN) during the later stages of training on this difficult task ([loss curves](https://imgur.com/a/bhoBfsw)) ; we therefore evaluated its best-performing checkpoint (based on pseudo-Dice). In contrast, TACO demonstrated superior training stability and achieved substantial performance gains:
> > >   - Dice: TACO (0.6114) vs. From-scratch (0.4119) and MAE (0.3430).
> > >   - NSD: TACO (0.8133) vs. From-scratch (0.5530) and MAE (0.4708).
> > > - Statistical Significance: To formally address the reviewer's concern, we conducted paired t-tests. The results confirm that TACO’s improvements are statistically significant in this task:
> > >   - Ours vs. from-scratch: Dice ($p = 0.0342$), NSD ($p = 0.0425$).
> > >   - Ours vs. MAE: Dice ($p = 0.0024$), NSD ($p = 0.0024$).
> > >   - In contrast, there was no significant difference between the MAE and from-scratch baselines ($p > 0.4$), highlighting that our method provides a unique advantage.
> > >
> > > As suggested, we have included these new results and significance analyses in the final manuscript. We believe this evidence provides a robust validation of our method's efficacy and its ability to generalize to challenging clinical scenarios. In addition, we plan to further explore the potential of our framework in more diverse and challenging clinical scenarios.
> > >
> > > [1] Timmins K M, van der Schaaf I C, Bennink E, et al. Comparing methods of detecting and segmenting unruptured intracranial aneurysms on TOF-MRAS: the ADAM challenge[J]. Neuroimage, 2021, 238: 118216.

---

### Official Review · Reviewer_rbZy · 2026-03-09

**Soundness:** 3
**Presentation:** 3
**Significance:** 2
**Originality:** 2
**Overall Recommendation:** 4
**Confidence:** 3

**Summary:**

This paper proposes a self-supervised pre-training framework (TACO) for multimodal medical imaging pretraining, that goes beyond instance-level SSL by considering cross-instance topological consistency. The proposed approach proposes an intra-instance cross-modal triplet loss, which uses registered modalities within the same subject, and an inter-instance loss using mutual nearest neighbor pseudo-correspondences across subjects. The experiments evaluate four segmentation tasks, two classification tasks, and a missing-modality segmentation task, showing improved average performance over several medical SSL baselines.

**Compliance With Llm Reviewing Policy:**

Affirmed.

**Final Justification:**

My concerns have been mostly resolved. Authors are encouraged to include the additional experimental results in their final version.

**Key Questions For Authors:**

Please see the weakness section above. My main concerns are about the missing discussion on the latest development on SSL, and the lack of ML cores in this work.

**Limitations:**

yes

**Strengths And Weaknesses:**

Strengths:
- The proposed idea that anatomy exhibits stable population-level spatial organization and can be used as supervision beyond instance-level SSL, is interesting and also well motivated for medical imaging.
- The presentation is clean and clear, visualizations on embeddings are insightful. Experiments are extensive.

Weaknesses:
- Recent topics and works on SSL medical imaging are not discussed. For example, in the SSL3D OpenMind (https://github.com/MIC-DKFZ/ssl3d) and FOMO (https://fomo25.github.io/), SSL models have been extensively discussed.
- Multimodal pre-training has also been explored a lot, which is not discussed sufficiently. There are open-source tools that can be used without fine-tuning, for example, FreeSurfer SuperSynth (https://surfer.nmr.mgh.harvard.edu/fswiki/SuperSynth) which can handle multimodal inputs and support fundamental medical imaging tasks.
- Following on the point above, both OpenMind and FOMO from 2025 have released large-scale datasets for SSL pre-training. Benchmarking on these datasets and their downstream applications and comparing with SOTA from these challenges will make this proposed framework more convincing.
- Overall, I think this work is more of a CV paper than an ML paper. This is not exactly a weakness, but I don't see it's a good fit for ICML and its general audience.

---

> ### Author Rebuttal · Authors · 2026-03-31
>
> Thank you for the constructive comments. Our responses are provided below.
> ### W1 & W2 & W3: Lack of discussion on OpenMind, FOMO, SuperSynth, and FreeSurfer. Benchmarking on these datasets and their downstream applications.
>
> We sincerely appreciate the reviewer’s constructive feedback regarding the latest methods. We will revise our manuscript to extensively discuss these landmark works: (1) large-scale curated dataset OpenMind and FOMO, (2) Downstream task tools SuperSynth and FreeSurfer.
>
> **SSL Benchmark Expansion:** To ensure a rigorous evaluation, we integrated our framework into the SSL3D OpenMind ecosystem during the rebuttal phase. Due to time and data access constraints, we evaluated TACO against these SOTA baselines across two segmentation datasets, ISLES and MSLS2017. For a fair and head-to-head comparison, we benchmarked against the top-performing model, MAE, in the OpenMind suite for segmentation.  We employed their officially released pre-trained weights and strictly followed the standardized nnU-Net v2 5-fold cross-validation and official downstream protocols. The reported evaluation metrics are DSCO(Dice Similarity Coefficient) and NSD(Normalized Surface Distance). We provide both average and the standard deviation value.
>
> As the following table shows, TACO achieves a 1.32% DSC gain on ISLES and a 0.84% NSD gain on MSLS2027, demonstrating superior segmentation and boundary alignment.
> | Pre-training method | ISLES DSC Avg. | ISLES DSC Std. | ISLES NSD Avg. | ISLES NSD Std. | MSLS2017 DSC Avg. | MSLS2017 DSC Std. | MSLS2017 NSD Avg. | MSLS2017 NSD Std. |
> |---|---:|---:|---:|---:|---:|---:|---:|---:|
> | MAE (best in OpenMind) | 74.02 | 2.38 | 71.82 | 0.85 | 83.02 | 4.46 | 92.66 | 4.82 |
> | TACO (Ours) | 75.34 | 2.44 | 72.12 | 1.28 | 83.96 | 4.50 | 93.50 | 4.81 |
>
> Furthermore, we plan to extend our method to the FOMO300K dataset and will release our models pre-trained on OpenMind114K and FOMO300K upon publication.
>
> **SuperSynth & FreeSurfer:** We thank the reviewer for this valuable suggestion. We agree that the discussion of prior multimodal pre-training and related open-source tools should be strengthened, and we will revise the manuscript accordingly.
>
> Regarding the examples mentioned, we would like to clarify that SuperSynth and FreeSurfer are highly useful neuroimaging tools, but they are not directly comparable to the setting considered in our work. SuperSynth is built upon BrainFM [1], a modality-agnostic multi-task supervised framework that leverages downstream task annotations during training. In contrast, our paper focuses on a pretraining paradigm without using downstream task labels, which addresses a different methodological question.
>
> FreeSurfer, on the other hand, is primarily a neuroimaging software suite for brain MRI processing, analysis, and visualization, rather than a general multimodal pretraining framework. While it supports important brain imaging workflows, its objective and usage scenario differ substantially from the representation learning setting studied in our paper.
>
> We appreciate the reviewer for pointing out these relevant tools. In the revision, we will explicitly discuss them to better position our work with respect to existing multimodal neuroimaging methods and toolkits.
>
> [1] A Modality-agnostic Multi-task Foundation Model for Human Brain Imaging
>
> ### W4: Relation to Machine Learning.
> Even though our method is developed for self-supervised learning in medical imaging, it relies on a nontrivial geometric assumption. As discussed in the paper, human anatomy exhibits a largely consistent spatial organization, which suggests an underlying homeomorphic structure. Based on this intuition, we assume the existence of a continuous map $C_{i\rightarrow j}^{h\rightarrow g}: \mathbb{R}^n \rightarrow \mathbb{R}^n$ between $Z_i^h$ and $Z_j^g$. Since all the latent representations are in $\mathbb{R}^n$ space, which is contractible, we have $C_{i\rightarrow j}^{h\rightarrow g} \simeq ID$, where $ID$ is the identity mapping.  Since $\sigma_{i,h} = \sigma_{j,g}\circ C_{i\rightarrow j}^{h\rightarrow g}$, it implies that $\sigma_{i,h}$ and $\sigma_{j,g}$ belong to the same homotopy class. So, the IM-NRC assumption indicates that the model preserves structurally consistent semantic information as long as the underlying topology, such as connectivity, is maintained, thereby achieving the goal of cross-modal and cross-instance alignment in self-supervised learning.
>
> We believe this perspective extends well beyond the scope of computer vision and is closely connected to broader problems in machine learning, including invariant representation learning, manifold alignment, etc. We will add this discussion to the revised paper. Meanwhile, our paper is submitted to the application track.

---

> > ### Author Rebuttal · Reviewer_rbZy · 2026-04-03
> >
> > Thanks for your clarification. My concerns have been mostly resolved. Authors are encouraged to include the additional experimental results in their final version.

---

> > > ### Author Response · Authors · 2026-04-08
> > >
> > > We thank the reviewer for the positive feedback. As suggested, we have integrated all additional experimental results and statistical analyses into the final manuscript. We are pleased that our responses and the new evidence have resolved the previous concerns.

---

### Decision · Program_Chairs · 2026-04-30

**Decision:**

Accept (regular)

**Comment:**

The paper presents a self-supervised strategy that leverages anatomic consistency to improve the learned representation. The proposed method introduces two training objectives: i) a triplet loss that preserves local neighborhoods between different (aligned) modalities of the same patient, and ii) an inter-class loss, which first finds correspondences between patches from different patients and then performs a similar triplet loss.

Initially, two reviewers provided a positive score for the paper, while one provided a weak reject.
Rev. rbZy appreciated the clean and clear presentation, and the use of anatomical consistency is sensible. However, they found the related work incomplete for semi-supervised learning, but also for multi-modal pre-training. Also, new datasets have been presented and should be used.
Rev. EeFS also appreciated the clean and clear presentation of the paper and the clear skins. They had concerns about technical details.
Rev. SRWu found the paper well written and clear, with the use of anatomic consistency and a compelling evaluation on several datasets. They had concerns about the structure of the manuscript and the validity of some experiments.

The authors wrote an excellent rebuttal that satisfied all authors' questions, and changed rev. EeFS opinion to weak accept.  After the rebuttal, all issues were closed. The only remaining issue is the improvement in the results, which look limited for certain.
Overall, I consider the contribution of the paper important, well presented, and useful, and I will recommend it for acceptance.